# Neutralizing Self-Selection Bias in Sampling for Sortition

**Bailey Flanigan,**
Computer Science Department
Carnegie Mellon University

**Paul Gölz**
Computer Science Department
Carnegie Mellon University

**Anupam Gupta**
Computer Science Department
Carnegie Mellon University

**Ariel D. Procaccia**
School of Engineering and Applied Sciences
Harvard University

## Abstract

Sortition is a political system in which decisions are made by panels of randomly selected citizens. The process for selecting a sortition panel is traditionally thought of as uniform sampling without replacement, which has strong fairness properties. In practice, however, sampling without replacement is not possible since only a fraction of agents is willing to participate in a panel when invited, and different demographic groups participate at different rates. In order to still produce panels whose composition resembles that of the population, we develop a sampling algorithm that restores close-to-equal representation probabilities for all agents while satisfying meaningful demographic quotas. As part of its input, our algorithm requires probabilities indicating how likely each volunteer in the pool was to participate. Since these participation probabilities are not directly observable, we show how to learn them, and demonstrate our approach using data on a real sortition panel combined with information on the general population in the form of publicly available survey data.

## 1  Introduction

What if political decisions were made not by elected politicians but by a randomly selected panel of citizens? This is the core idea behind *sortition*, a political system originating in the Athenian democracy of the 5th century BC [24]. A *sortition panel* is a randomly selected set of individuals who are appointed to make a decision on behalf of population from which they were drawn. Ideally, sortition panels are selected via uniform sampling without replacement — that is, if a panel of size $k$ is selected from a population of size $n$, then each member of the population has a $k/n$ probability of being selected. This system offers appealing fairness properties for both individuals and subgroups of the population: First, each individual knows that she has the same probability of being selected as anyone else, which assures her an equal say in decision making. The resulting panel is also, in expectation, *proportionally representative* to all groups in the population: if a group comprises $x\%$ of the population, they will in expectation comprise $x\%$ of the panel as well. In fact, if $k$ is large enough, concentration of measure makes it likely that even a group's *ex post* share of the panel will be close to $x\%$. Both properties stand in contrast to the status quo of electoral democracy, in which the equal influence of individuals and the fair participation of minority groups are often questioned.

Due to the evident fairness properties of selecting decision makers randomly, sortition has seen a recent surge in popularity around the world. Over the past year, we have spoken with several nonprofit organizations whose role it is to sample and facilitate sortition panels [8]. One of these nonprofits, the

*Sortition Foundation*, has organized more than 20 panels in about the past year.[1] Recent high-profile examples of sortition include the Irish Citizens' Assembly,[2] which led to Ireland's legalization of abortion in 2018, and the founding of the first permanent sortition chamber of government,[3] which occurred in a regional parliament in the German-speaking community of Belgium in 2019.

The fairness properties of sortition are often presented as we have described them — in the setting where panels are selected *from the whole population* via uniform sampling without replacement. As we have learned from practitioners, however, this sampling approach is not applicable in practice due to limited participation: typically, only between 2 and 5% of citizens are willing to participate in the panel when contacted. Moreover, those who do participate exhibit self-selection bias, i.e., they are not representative of the population, but rather skew toward certain groups with certain features.

To address these issues, sortition practitioners introduce additional steps into the sampling process. Initially, they send a large number of invitation letters to a random subset of the population. If a recipient is willing to participate in a panel, they can opt into a *pool* of volunteers. Ultimately, the panel of size $k$ is sampled from the pool. Naturally, the pool is unlikely to be representative of the population, which means that uniformly sampling from the pool would yield panels whose demographic composition is unrepresentative of that of the population. To prevent grossly unrepresentative panels, many practitioners impose quotas on groups based on orthogonal demographic features such as gender, age, or residence inside the country. These quotas ensure that the ex-post number of panel members belonging to such a group lies within a narrow interval around the proportional share. Since it is hard to construct panels satisfying a set of quotas, practitioners typically sample using greedy heuristics. One downside of these greedy algorithms is that they may need many attempts to find a valid panel, and thus might take exponential time to produce a valid panel if one exists. More importantly, however, even though these heuristics tend to eventually find valid panels, the probability with which each individual is selected via their selection process is not controlled in a principled way.

By not deliberately controlling individual selection probabilities, existing panel selection procedures fail to achieve basic fairness guarantees to individuals. Where uniform sampling in the absence of self-selection bias selects each person with equal probability $k/n$, currently-used greedy algorithms do not even guarantee a minimum selection probability for members of the *pool*, let alone fairly distributed probabilities over members of the population. The absence of such a theoretical guarantee has real ramifications in practice: as we show in Section 5, in the real-world instance we study, the greedy algorithm selects several population members with probability less than half the magnitude of $k/n$.

This unfairness to individuals, while problematic in its own right, can also lead to unfairness for groups. In particular, current algorithms use quotas to enforce representation of a set of pre-chosen groups delineated by single features, but these quotas do not protect groups defined by *intersections* of these features: for example, proportional representation of women and of young people does not guarantee the proportional representation of young women. By giving some individuals very low probability of selection on the basis of their *combinations* of features, existing algorithms may systematically allocate very low probability to all members of a certain intersectional group, preventing their perspectives from being fairly represented on the panel.

## 1.1 Our Techniques and Results

The main contribution of this paper is a more principled sampling algorithm that, even in the presence of limited participation and self-selection bias, retains the individual fairness properties of uniform sampling in the absence of these challenges, while also allowing the deterministic satisfaction of quotas. In particular, our algorithm satisfies the following desiderata:

- *End-to-End Fairness:* The algorithm selects the panel via a process such that all members of the population appear on the panel with probability asymptotically close to $k/n$. This also implies that all groups in the population, including those defined by intersections of arbitrarily many features, will be near-proportionally represented in expectation.
- *Deterministic Quota Satisfaction:* The selected panel satisfies certain upper and lower quotas enforcing approximate representation for a set of specified features.

– *Computational Efficiency:* The algorithm returns a valid panel (or fails) in polynomial time.

*End-to-end* fairness refers to the fact that our algorithm is fair to individuals with respect to their probabilities of going from *population* to *panel*, across the intermediate steps of being invited, opting into the pool, and being selected for the panel. End-to-end fairness can be seen primarily as a guarantee of individual fairness, while proportional representation of all groups in expectation, along with deterministic quota satisfaction, can be seen as two different guarantees of group fairness.

The key challenge in satisfying these desiderata is self-selection bias, which can result in the pool being totally unrepresentative of the population. In the worst case, the pool can be so skewed that it contains no representative panel — in fact, the pool might not even contain $k$ members. As a result, no algorithm can produce a valid panel from every possible pool. However, we are able to give an algorithm that succeeds with high probability, under weak assumptions mainly relating the number of invitation letters sent out to $k$ and the minimum participation probability over all agents.

Crucially, any sampling algorithm that gives (near-)equal selection probability to all members of the population must reverse the self-selection bias occurring in the formation of the pool. We formalize this self-selection bias by assuming that each agent $i$ in the population agrees to join the pool with some positive participation probability $q_i$ when invited. If these $q_i$ values are known for all members of the pool, our sampling algorithm can use them to neutralize self-selection bias. To do so, our algorithm selects agent $i$ for the panel with a probability (close to) proportional to $1/q_i$, conditioned on $i$ being in the pool. This compensates for agents' differing likelihoods of entering the pool, thereby giving all agents an equal end-to-end probability. On a given pool, the algorithm assigns marginal selection probabilities to every agent in the pool. Then, to find a distribution over valid panels that implements these marginals, the algorithm randomly rounds a linear program using techniques based on discrepancy theory. Since our approach aims for a fair *distribution* of valid panels rather than just a single panel, we can give probabilistic fairness guarantees.

As we mentioned, our theoretical and algorithmic results, presented in Section 3, take the probabilities $q_i$ of all pool members $i$ as given in the input. While these values are not observed in practice, we show in Section 4 that they can be estimated from available data. We cannot directly train a classifier predicting participation, however, because practitioners collect data only on those who *do* join the pool, yielding only positively labeled data. In place of a negatively labeled control group, we use publicly available survey data, which is unlabeled (i.e., includes no information on whether its members would have joined the pool). To learn in this more challenging setting, we use techniques from *contaminated controls*, which combine the pool data with the unlabeled sample of the population to learn a predictive model for agents' participation probabilities. In Section 5, we use data from a real-world sortition panel to show that plausible participation probabilities can be learned and that the algorithm produces panels that are close to proportional across features. For a synthetic population produced by extrapolating the real data, we show that our algorithm obtains fair end-to-end probabilities.

## 1.2 Related Work

Our work is broadly related to existing literature on fairness in the areas of *machine learning*, *statistics*, and *social choice*. Through the lens of fair machine learning, our quotas can be seen as enforcing approximate statistical fairness for protected groups, and our near-equal selection probability as a guarantee on individual fairness. Achieving simultaneous group- and individual-level fairness is a commonly discussed goal in fair machine learning [5, 12, 15], but one that has proven somewhat elusive. To satisfy fairness constraints on orthogonal protected groups, we draw upon techniques from discrepancy theory [2, 3], which we hope to be more widely applicable in this area.

Our paper addresses self-selection bias, which is routinely faced in statistics and usually addressed by sample reweighting. Indeed, our sampling algorithm can be seen as a way of reweighting the pool members under the constraint that weights must correspond to the marginal probabilities of a random distribution. While reweighting is typically done by the simpler methods of post-stratification, calibration [14], and sometimes regression [20], we use the more powerful tool of learning with contaminated controls [16, 27] to determine weights on a more fine-grained level.

Our paper can also be seen as a part of a broader movement towards statistical approaches in social choice [17, 18, 22]. The problem of selecting a representative sortition panel can be seen as a fair division problem, in which $k$ indivisible copies of a scarce resource must be randomly allocated such

that an approximate version of the proportionality axiom is imposed. Our group fairness guarantees closely resemble the goal of apportionment, in which seats on a legislature are allocated to districts or parties such that each district is proportionally represented within upper and lower quotas [1, 6, 13].

So far, only few papers in computer science and statistics directly address sortition [4, 21, 26]. Only one of them [4] considers, like us, how to sample a representative sortition panel. Unfortunately, their stratified sampling algorithm assumes that all agents are willing to participate, which, as we address in this paper, does not hold in practice.

## 2 Model

**Agents.** Let $N$ be a set of $n$ agents, constituting the underlying population. Let $F$ be a set of *features*, where feature $f \in F$ is a function $f : N \to V_f$, mapping the agents to a set $V_f$ of possible values of feature $f$. For example, for the feature *gender*, we could have $V_{gender} = \{male, female, non-binary\}$. Let the *feature-value pairs* be $\bigcup_{f \in F}\{(f, v) \mid v \in V_f\}$. In our example, the feature-value pairs are $(gender, male)$, $(gender, female)$, and $(gender, non-binary)$. Denote the number of agents with a particular feature-value pair $(f, v)$ by $n_{f,v}$.

Each agent $i \in N$ is described by her *feature vector* $F(i) := \{(f, f(i)) \mid f \in F\}$, the set of all feature-value pairs pertaining to this agent. Building on the example instance, suppose we add the feature *education-level*, so $F = \{gender, education\ level\}$. If *education level* can take on the values *college* and *no college*, a college-educated woman would have the feature-vector $\{(gender, female), (education\ level, college)\}$.

**Panel Selection Process.** Before starting the selection process, organizers of a sortition panel must commit to the panel's parameters. First, they must choose the number of *recipients* $r$ who will be invited to potentially join the panel, and the required *panel size* $k$. Moreover, they must choose a set of features $F$ and values $\{V_f\}_{f \in F}$ over which quotas will be imposed. Finally, for all feature-value pairs $(f, v)$, they must choose a *lower quota* $\ell_{f,v}$ and an *upper quota* $u_{f,v}$, implying that the eventual panel of $k$ agents must contain *at least* $\ell_{f,v}$ and *at most* $u_{f,v}$ agents with value $v$ for feature $f$. Once these parameters are fixed, the panel selection process proceeds in three steps:

$$population \xrightarrow{\textbf{STEP 1}} recipients \xrightarrow{\textbf{STEP 2}} pool \xrightarrow{\textbf{STEP 3}} panel$$

In **STEP 1**, the organizer of the panel sends out $r$ letters, inviting a subset of the population — sampled with equal probability and without replacement — to volunteer for serving on the panel. We refer to the random set of agents who receive these letters as $Recipients$. Only the agents in $Recipients$ will have the opportunity to advance in the process toward being on the panel.

In **STEP 2**, each letter recipient may respond affirmatively to the invitation, thereby opting into the pool of agents from which the panel will be chosen. These agents form the random set $Pool$, defined as the set of agents who received a letter and agreed to serve on the panel if ultimately chosen. We assume that each agent $i$ joins the pool with some *participation probability* $q_i > 0$. Let $q^*$ be the lowest value of $q_i$ across all agents $i \in N$. A key parameter of an instance is $\alpha := q^* r/k$, which measures how large the number of recipients is relative to the other parameters. Larger values of $\alpha$ will allow us the flexibility to satisfy stricter quotas.

In **STEP 3**, the panel organizer runs a *sampling algorithm*, which selects the panel from the pool. This panel, denoted as the set $Panel$, must be of size $k$ and satisfy the predetermined quotas for all feature-value pairs. The sampling algorithm may also fail without producing a panel.

We consider the first two steps of the process to be fully prescribed. The focus of this paper is to develop a sampling algorithm for the third step that satisfies the three desiderata listed in the introduction: end-to-end fairness, deterministic quota satisfaction, and computational efficiency.

## 3 Sampling Algorithm

In this section, we give an algorithm which ensures, under natural assumptions, that every agent ends up on the panel with probability at least $(1 - o(1))\, k/n$ as $n$ goes to infinity.[4] Furthermore, the panels

produced by this algorithm satisfy non-trivial quotas, which ensure that the ex-post representation of each feature-value pair cannot be too far from being proportional.

Our algorithm proceeds in two phases: *I. assignment of marginals*, during which the algorithm assigns a marginal selection probability to every agent in the pool, and *II. rounding of marginals*, in which the marginals are dependently rounded to $0/1$ values, the agents' indicators of being chosen for the panel. As we discussed previously, our algorithm succeeds only with high probability, rather than deterministically; it may fail in phase I if the desired marginals do not satisfy certain conditions. We refer to pools on which our algorithm succeeds as *good pools*. A good pool, to be defined precisely later, is one that is highly representative of the population — that is, its size and the prevalence of all feature values within it are close to their respective expected values. We leave the behavior of our algorithm on bad pools unspecified: while the algorithm may try its utmost on these pools, we give no guarantees in these cases, so the probability of representation guaranteed to each agent must come only from good pools and valid panels. Fortunately, under reasonable conditions, we show that the pool will be good with high probability. When the pool is good, our algorithm always succeeds, meaning that our algorithm is successful overall with high probability.

Our algorithm satisfies the following theorem, guaranteeing close-to-equal end-to-end selection probabilities for all members of the population as well as the satisfaction of quotas.

**Theorem 1.** *Suppose that $\alpha \to \infty$ and $n_{f,v} \geq n/k$ for all feature-value pairs $f, v$. Consider a sampling algorithm that, on a good pool, selects a random panel, $Panel$, via the randomized version of Lemma 3, and else does not return a panel. This process satisfies, for all $i$ in the population, that*

$$\mathbb{P}[i \in Panel] \geq (1 - o(1))\, k/n.$$

*All panels produced by this process satisfy the quotas $\ell_{f,v} := (1 - \alpha^{-.49})\, k\, n_{f,v}/n - |F|$ and $u_{f,v} := (1 + \alpha^{-.49})\, k\, n_{f,v}/n + |F|$ for all feature-value pairs $f, v$.*

The guarantees of the theorem grow stronger as the parameter $\alpha = q^* r/k$ tends toward infinity, i.e., as the number $r$ of invitations grows. Note that, since $r \leq n$, this assumption requires that $q^* \gg k/n$. We defer all proofs to Appendix B and discuss the preconditions in Appendix B.1.

## 3.1 Algorithm Part I: Assignment of Marginals

To afford equal probability of panel membership to each agent $i$, we would like to select agent $i$ with probability inversely proportional to her probability $q_i$ of being in the pool. For ease of notation, let $a_i := 1/q_i$ for all $i$. Specifically, for agent $i$, we want $\mathbb{P}[i \in Panel \mid i \in Pool]$ to be proportional to $a_i$. Achieving this exactly is tricky, however, because each agent's *selection probability* from pool $P$, call it $\pi_{i,P}$, must depend on those of all other agents in the pool, since their marginals must add to the panel size $k$. Thus, instead of reasoning about an agent's probability across all possible pools at once, we take the simpler route of setting agents' selection probabilities for each pool separately, guaranteeing that $\mathbb{P}[i \in Panel \mid i \in P]$ is proportional to $a_i$ across all members $i$ of a good pool $P$. For any good pool $P$, we select each agent $i \in P$ for the panel with probability

$$\pi_{i,P} := k\, a_i / \sum_{j \in P} a_j.$$

Note that this choice ensures that the marginals always sum up to $k$.

**Definition of Good Pools.** For this choice of marginals to be reasonable and useful for giving end-to-end guarantees, the pool $P$ must satisfy three conditions, whose satisfaction defines a *good pool $P$*. First, the marginals do not make much sense unless all $\pi_{i,P}$ lie in $[0, 1]$:

$$0 \leq \pi_{i,P} \leq 1 \quad \forall i \in P. \tag{1}$$

Second, the marginals summed up over all pool members of a feature-value pair $f, v$ should not deviate too far from the proportional share of the pair:

$$(1 - \alpha^{-.49})\, k\, n_{f,v}/n \leq \sum_{i \in P: f(i)=v} \pi_{i,P} \leq (1 + \alpha^{-.49})\, k\, n_{f,v}/n \quad \forall f, v. \tag{2}$$

Third, we also require that the term $\sum_{i \in P} a_i$ is not much larger than $\mathbb{E}[\sum_{i \in Pool} a_i] = r$, which ensures that the $\pi_{i,P}$ do not become to small:

$$\sum_{i \in P} a_i \leq r/(1 - \alpha^{-.49}). \tag{3}$$

Under the assumptions of our theorem, pools are good with high probability, even if we condition on any agent $i$ being in the pool:

**Lemma 2.** *Suppose that $\alpha \to \infty$ and $n_{f,v} \geq n/k$ for all $f, v$. Then, for all agents $i \in Population$, $\mathbb{P}[Pool$ is good $\mid i \in Pool] \to 1$.*

Note that only constraint (1) prevents Phase II of the algorithm from running; the other two constraints just make the resulting distribution less useful for our proofs. In practice, if it is possible to rescale the $\pi_{i,P}$ and cap them at 1 such that their sum is $k$, running phase II on these marginals seems reasonable.

### 3.2 Algorithm Part II: Rounding of Marginals

The proof of Theorem 1 now hinges on our ability to implement the chosen $\pi_{i,P}$ for a good pool $P$ as marginals of a distribution over panels. This phase can be expressed in the language of randomized dependent rounding: we need to define random variables $X_i = \mathbb{1}\{i \in Panel\}$ for each $i \in Pool$ such that $\mathbb{E}[X_i] = \pi_{i,P}$. This difficulty of this task stems from the ex-post requirements on the pool, which require that $\sum_i X_i = k$ and that $\sum_{i:f(i)=v} X_i$ is close to $k\,n_{f,v}/n$ for all feature-value pairs $f, v$. While off-the-shelf dependent rounding [10] can guarantee the marginals and the sum-to-$k$ constraint, it cannot simultaneously ensure small deviations in terms of the representation of all $f, v$.

Our algorithm uses an iterative rounding procedure based on a celebrated theorem by Beck and Fiala [3]. We sketch here how to obtain a deterministic rounding satisfying the ex-post constraints; the argument can be randomized using results by Bansal [2] or via column generation (Appendix B.4.2).[5] The iterated rounding procedure manages a variable $x_i \in [0, 1]$ for each $i \in Pool$, which is initialized as $\pi_{i,P}$. As the $x_i$ are repeatedly updated, more of them are fixed as either 0 or 1 until the $x_i$ ultimately correspond to indicator variables of a panel. Throughout the rounding procedure, it is preserved that $\sum_i x_i = \sum_i \pi_{i,P} = k$, and the equalities $\sum_{i:f(i)=v} x_i = \sum_{i:f(i)=v} \pi_{i,P}$ are preserved until at most $|F|$ variables $x_i$ in the sum are yet to be fixed. As a result, the final panel has exactly $k$ members, and the number of members from a feature-value pair $f, v$ is at least $\sum_{i:f(i)=v} \pi_{i,P} - |F| \geq (1 - \alpha^{-.49})\,k\,n_{f,v}/n - |F|$ (symmetrically for the upper bound).[6] As we show in Appendix B.4,

**Lemma 3.** *There is a polynomial-time sampling algorithm that, given a good pool $P$, produces a random panel Panel such that (1) $\mathbb{P}[i \in Panel] = \pi_{i,P}$ for all $i \in P$, (2) $|Panel| = k$, and (3) $\sum_{i:f(i)=v} \pi_{i,P} - |F| \leq |\{i \in Panel \mid f(i) = v\}| \leq \sum_{i:f(i)=v} \pi_{i,P} + |F|$.*

Our main theorem follows from a simple argument combining Lemmas 2 and 3 (Appendix B.5).

While the statement of Theorem 1 is asymptotic in the growth of $\alpha$, the same proof gives bounds on the end-to-end probabilities for finite values of $\alpha$. If one wants bounds for a specific instance, however, bounds uniquely in terms of $\alpha$ tend to be loose, and one might want to relax Condition (2) of a good pool in exchange for more equal end-to-end probabilities. In this case, plugging the specific values of $n, r, k, q^*, n_{f,v}$ into the proof allows to make better trade-offs and to extract sharper bounds.

## 4 Learning Participation Probabilities

The algorithm presented in the previous section relies on knowing $q_i$ for all agents $i$ in the pool. While these $q_i$ are not directly observed, we can estimate them from data available to practitioners.

First, we assume that an agent $i$'s participation probability $q_i$ is a function of her feature vector $F(i)$. Furthermore, we assume that $i$ makes her decision to participate through a specific generative model known as *simple independent action* [11, as cited in [28]]. First, she flips a coin with probability $\beta_0$ of landing on heads. Then, she flips a coin for each feature $f \in F$, where her coin pertaining to $f$ lands on heads with probability $\beta_{f,f(i)}$. She participates in the pool if and only if all coins she flips land on heads, leading to the following functional dependency:

$$q_i = \beta_0 \prod_{f \in F} \beta_{f,f(i)}.$$

We think of $1 - \beta_{f,v}$ as the probability that a reason specific to the feature-value pair $f, v$ prevents the agent from participating, and of $1 - \beta_0$ as the baseline probability of her not participating for reasons independent of her features. The simple independent action model assumes that these reasons occur independently between features, and that the agent participates iff none of the reasons occur.

If we had a representative sample of agents — say, the recipients of the invitation letters — labeled according to whether they decided participate ("positive") or not ("negative"), learning the parameters $\beta$ would be straightforward. However, sortition practitioners only have access to the features of those who enter the pool, and not of those who never respond. Without a control group, it is impossible to distinguish a feature that is prevalent in the population and associated with low participation rate from a rare feature associated with a high participation rate. Thankfully, we can use additional information: in place of a negatively-labeled control group, we use a *background sample* — a dataset containing the features for a uniform sample of agents, but without labels indicating whether they would participate. Since this control group contains both positives and negatives, this setting is known as *contaminated controls*. A final piece of information we use for learning is the fraction $\overline{q} := |Pool|/r$, which estimates the mean participation probability across the population. In other applications with contaminated controls, including $\overline{q}$ in the estimation increased model identifiability [27].

To learn our model, we apply methods for maximum likelihood estimation (MLE) with contaminated controls introduced by Lancaster and Imbens [16]. By reformulating the simple independent action model in terms of the logarithms of the $\beta$ parameters, their estimation (with a fixed value of $\overline{q}$) reduces to maximizing a concave function.

**Theorem 4.** *The log-likelihood function for the simple independent action model under contaminated controls is concave in the model parameters.*

By this theorem, proven in Appendix C, we can directly and efficiently estimate $\beta$. Logistic models, by contrast, require more involved techniques for efficient estimation [27].

## 5   Experiments

**Data.** We validate our $q_i$ estimation and sampling algorithm on pool data from *Climate Assembly UK*,[7] a national-level sortition panel organized by the Sortition Foundation in 2020. The panel consisted of $k = 110$ many UK residents aged 16 and above. The Sortition Foundation invited all members of $30\,000$ randomly selected households, which reached an estimated $r = 60\,000$ eligible participants.[8] Of these letter recipients, $1\,715$ participated in the pool,[9] corresponding to a mean participation probability of $\overline{q} \approx 2.9\%$. The feature-value pairs used for this panel can be read off the axis of Fig. 1. We omit an additional feature *climate concern level* in our main analysis because only 4 members of the pool have the value *not at all concerned*, whereas this feature-value pair's proportional number of panel seats is 6.5. To allow for proportional representation of groups with such low participation rates, $r$ should have been chosen to be much larger. We believe that the merits of our algorithm can be better observed in parameter ranges in which proportionality can be achieved. For the background sample, we used the 2016 European Social Survey [19], which contains $1\,915$ eligible individuals, all with features and values matching those from the panel. Our implementation is based on PyTorch and Gurobi, runs on consumer hardware, and its code is available on github. Appendix D contains details on Climate Assembly UK, data processing, the implementation, and further experiments (including the climate concern feature).

**Estimation of $\beta$ Parameters.** We find that the baseline probability of participation is $\beta_0 = 8.8\%$. Our $\beta_{f,v}$ estimates suggest that (from strongest to weakest effect) highly educated, older, urban, male, and non-white agents participate at higher rates. These trends reflect these groups' respective levels of representation in the pool compared to the underlying population, suggesting that our estimated $\beta$ values fit our data well. Different values of the remaining feature, region of residence, seem to have heterogeneous effects on participation, where being a resident of the South West gives substantially increased likelihood of participation compared to other areas. The lowest participation probability of any agent in the pool, according to these estimates, is $q^* = 0.78\%$, implying that $\alpha \approx 4.25$. See Appendix D.4 for detailed estimation results and validation.

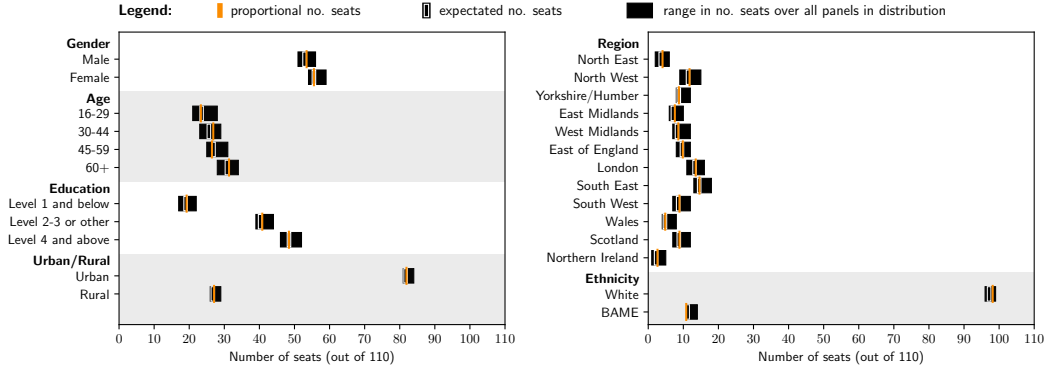

Figure 1: Expected and realized numbers of panel seats our algorithm gives each feature-value pair in the Climate Assembly pool.

**Running the Sampling Algorithm on the Pool.** The estimated $q_i$ allow us to run our algorithm on the Climate Assembly pool and thereby study its fairness properties for non-asymptotic input sizes. We find that the Climate Assembly pool is good relative to our $q_i$ estimates, i.e., that it satisfies Eqs. (1) to (3). As displayed in Fig. 1, the marginals produced by Phase I of our algorithm give each feature-value pair $f, v$ an expected number of seats, $\sum_{i \in P, f(i)=v} \pi_{i,P}$, within *one seat* of its proportional share of the panel, $k\,n_{f,v}/n$. By Lemma 3, Phase II of our algorithm then may produce panels from these marginals in which $f, v$ receives up to $|F| = 6$ fewer or more seats than its expected number. However, as the black bars in Fig. 1 show, the actual number of seats received by any $f, v$ across *any panel* produced by our algorithm on this input never deviates from its expectation by more than 4 seats. As a result, while Theorem 1 only implies lower quotas of $.51\,k\,n_{f,v}/n - |F|$ and upper quotas of $1.49\,k\,n_{f,v}/n + |F|$ for this instance, the shares of seats our algorithm produces lie in the much narrower range $k\,n_{f,v}/n \pm 5$ (and even $k\,n_{f,v}/n \pm 3$ for 18 out of 25 feature-value pairs). This suggests that, while the quotas guaranteed by our theoretical results are looser than the quotas typically set by practitioners, our algorithm will often produce substantially better ex-post representation than required by the quotas.

**End-to-End Probabilities.** In the previous experiments, we were only able to argue about the algorithm's behavior on a single pool. To validate our guarantees on individual end-to-end probabilities, we construct a synthetic population of size 60 million by duplicating the ESS participants, assuming our estimated $q_i$ as their true participation probabilities. Then, for various values of $r$, we sample a large number of pools. By computing $\pi_{i,P}$ values for all agents $i$ in each pool, we can estimate each agent's end-to-end probability of ending up on the panel. Crucially, we assume that our algorithm does not produce any panel for bad pools, analogously to Theorem 1. As shown in the following graph, for $r = 60\,000$ (as was used in Climate Assembly UK), all agents in our synthetic population, across the full range of $q_i$, receive probability within $.1\,k/n$ of $k/n$ (averaged over $100\,000$ random pools):

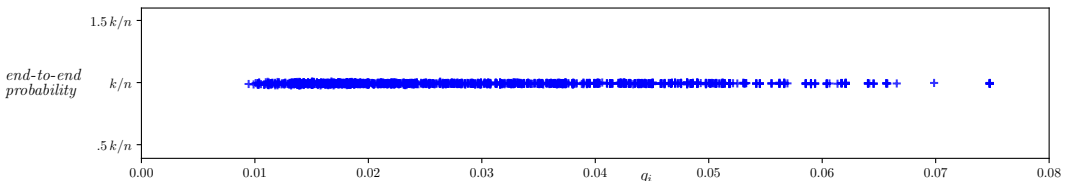

That these end-to-end probabilities are so close to $k/n$ also implies that bad pools are exceedingly rare for this value of $r$. As we show in Appendix D.6, we see essentially the same behavior for values of $r$ down to roughly $15\,000$, when $\alpha \approx 1$. For even lower $r$, most pools are bad, so end-to-end probabilities are close to zero under our premise that no panels are produced from bad pools.

To demonstrate that our algorithm's theoretical guarantees lead to realized improvements in individual fairness over the state-of-the-art, we re-run the experiment above, this time using the Sortition Foundation's greedy algorithm to select a panel from each generated pool. Since their algorithm requires explicit quotas as input, we set the lower and upper quotas for each feature-value group to

be the floor and ceiling of that group's proportional share of seats. This is a popular way of setting quotas in current practice.

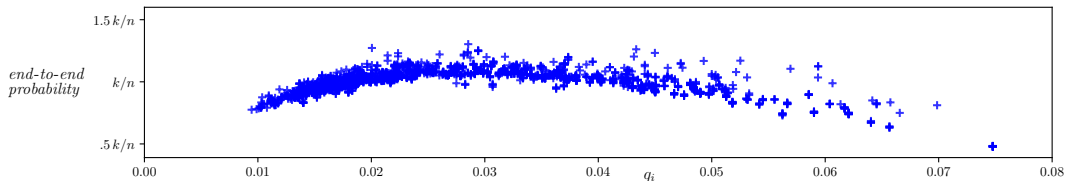

The results of this experiment show that the individual end-to-end probabilities generated by the currently-used greedy algorithm range from below $0.5\,k/n$ up to $1.3\,k/n$. In comparison to the end-to-end probabilities generated by our algorithm, those generated by the greedy algorithm are substantially skewed, and tend to disadvantage individuals with either low or high participation probabilities. One might argue that the comparison between our algorithm and the greedy is not quite fair, since the greedy algorithm is required to satisfy stronger quotas. However, looser quotas do not improve the behavior of the greedy algorithm; they simply make it behave more similarly to uniform sampling from the pool, which further disadvantages agents with low participation probability (for details, see Appendix D.5).

Taken together, these results illustrate that, although greedy algorithms like the one we examined achieve proportional representation of a few pre-specified groups via quotas, they do not achieve fairness to individuals or to groups unprotected by quotas. Compared to the naive solution of uniform sampling from the pool, greedily striving for quota satisfaction does lead to more equal end-to-end probabilities, as pool members with underrepresented features are more likely to be selected for the panel than pool members with overrepresented features. However, this effect does not neutralize self-selection bias when there are multiple features, even when selection bias acts through the independent-action model as in our simulated population. Indeed, in this experiment, the greedy algorithm insufficiently boosts the probabilities of agents in the intersection of multiple low-participation groups (the agents with lowest $q_i$), while also too heavily dampening the selection probability of those in the intersection of multiple high-participation groups (with highest $q_i$). These observations illustrate the need for panel selection algorithms that explicitly control individual probabilities.

## 6   Discussion

In a model in which agents *stochastically* decide whether to participate, our algorithm guarantees similar end-to-end probabilities to all members of the population. Arguably, an agent's decision to participate when invited might not be random, but rather *deterministically* predetermined.

From the point of view of such an agent $i$, does our algorithm, based on a model that doesn't accurately describe her (and her peers') behavior, still grant her individual fairness? If $i$ deterministically *participates*, the answer is yes (if not, of course she cannot be guaranteed anything). To see why, first observe that, insofar as it concerns $i$'s chance of ending up on the panel, all other agents might as well participate randomly.[10] Indeed, from agent $i$'s perspective, the process looks like the stochastic process where every other agent $j$ participates with probability $q_j$, where $i$ herself always participates, and where the algorithm erroneously assumes that $i$ joins only with some probability $q_i$. Therefore, the pool is still good with high probability conditioned on $i$ being in it, as argued in Lemma 2. Even if the algorithm knew that $q_i = 1$, $i$'s end-to-end probability would be at least $\bigl(1 - o(1)\bigr)\,k/n$, and the fact that the algorithm underestimates her $q_i$ only increases her probability of being selected from the pool. It follows that $i$'s end-to-end probability in this setting still must be at least around $k/n$.

Thus, in a deterministic model of participation, our individual guarantees are reminiscent of the axiom of population monotonicity in fair division: *If the whole population always participated when invited, every agent would reach the panel with probability $k/n$. The fact that some agents do not participate cannot (up to lower-order terms) decrease the selection probabilities for those who do.*

## Broader Impact

As we discussed in the paper, sortition is becoming increasingly widespread as a method for making collective decisions and gauging public opinion. Based on our experiences, practitioners seem interested in the possibility of fairer sampling algorithms, so our research in this area has the potential to influence how sortition panels are sampled in the real world.

From a **fairness** standpoint, our algorithm represents an improvement over currently-used algorithms on several fronts. We maintain the approximate satisfaction of proportional quotas while giving provable fairness guarantees to individuals, which in turn safeguard against systematic under-representation of demographic groups unprotected by quotas. Currently-used greedy algorithms give no such guarantees, and as we show in a real-world example, these theoretical differences can translate to substantial practical differences in the equality of individual selection probabilities. Our model is also robust to bias in the first stage of sampling: if a group is unfairly undersampled in the invitation stage, an effective machine-learning method will estimate lower $q_i$ for this group, and the algorithm will then correct this bias in the second stage. For these reasons, sortition panels selected via our algorithm will in general be more representative of the population, hopefully resulting in their decisions more fairly reflecting the interests and views of the entire population.

Of course, the methods we present do still come with fairness concerns. One main concern would be that the learned $q_i$ values could erroneously overestimate the participation probabilities of those in a certain group, thereby resulting in our sampling algorithm giving members of that group lower probability they deserve in the second step, and thereby giving them unfairly low end-to-end selection probability. We identify three main potential sources of inaccuracy in our MLE estimator: unreliable or unrepresentative background data, small sample sizes from which to learn, and failure of our model to accurately capture people's participation decisions. While each of these factors could potentially introduce inaccuracies in the $q_i$ values, it is unclear that any but the first issue is likely to result in systematic bias against certain groups.

Another important consideration in panel selection is **transparency** about the selection process to constituents on whose behalf the panel will make decisions. Increasing the transparency of how these panels are selected can increase public trust in the fairness of this method of decision-making, thus expanding sortition's legitimacy and reach. Toward the goal of transparency, our approach offers improvements on the state-of-the-art in several ways. First, this work gives formal theoretical guarantees of multiple types of fairness, where currently-used methods give none. Our algorithm also follows an explicit and explainable process for setting marginal selection probabilities, rather than having them accidentally arise from a greedy process.

Despite these improvements, our methods still present transparency challenges. Since an individual's probability of selection from the pool depends on their estimated $q_i$ value, the fairness of the process hinges on the entire machine-learning pipeline — data used, choice of model, and estimation methods — multiple elements of which might be opaque to most of the population. Secondly, while the use of protected attributes to counteract inequality is an accepted practice in the fairness in classification literature, there may still be public discomfort about this idea. In particular, it may be a source of discomfort that, in order to equalize end-to-end probabilities, our algorithm must explicitly decrease the probability of certain people being selected from the pool due to their attributes. One potentially comforting and easily-understood feature of our algorithm, however, is described in Section 6: that if someone participates in the pool with probability 1, they are guaranteed an end-to-end selection probability of at least $k/n$, regardless of their attributes.

## Acknowledgments and Disclosure of Funding

We thank Sivaraman Balakrishnan, Nikhil Bansal, and David Wajc for helpful technical discussions, and Terry Bouricious, Adam Cronkright, Linn Davis, Adela Gąsiorowska, Marcin Gerwin, Brett Hennig, David Schecter, and Robin Teater for sharing their insights on practical sortition. We would also like to express our gratitude to the Sortition Foundation for supplying the data used in our experiments.

This work was partially supported by the National Science Foundation under grants CCF-1907820, CCF1955785, CCF-2006953, CCF-1525932, CCF-1733556, CCF-2007080, and IIS-2024287; and by the Office of Naval Research under grant N00014-20-1-2488. Bailey Flanigan is supported by

the National Science Foundation Graduate Research Fellowship and the Fannie and John Hertz Foundation.

## Footnotes

[1] https://www.youtube.com/watch?v=hz2d_8eBEKg at 8:53.

[2] https://2016-2018.citizensassembly.ie/en/

[3] https://www.politico.eu/article/belgium-democratic-experiment-citizens-assembly/

[4] We allow $k \geq 1$ and $r \geq 1$ to vary arbitrarily in $n$ and assume that the feature-value pairs are fixed.

[5]Bansal [2] gives a black-box polynomial-time method for randomizing our rounding procedure. We found column-generation-based algorithms to be faster in practice, with guarantees that are at least as tight.

[6]Observe that our Beck-Fiala-based rounding procedure only increases the looseness of the quotas by a constant additive term beyond the losses to concentration. The concentration properties of standard dependent randomized rounding do not guarantee such a small gap with high probability. Moreover, our bound does not directly depend on the number of quotas (i.e., twice the number of feature-value pairs) but only depends on the number of features, which are often much fewer.

[7]https://www.climateassembly.uk/

[8]Note that every person in the population has equal probability $(30\,000/\#\text{households})$ of being invited. We ignore correlations between members of the same household.

[9]Excluding 12 participants with gender "other" as no equivalent value is present in the background data.

[10]Fix a group of agents who, assuming the stochastic model, will participate if invited with probability $q$. Then, sampling letter recipients from this set of agents in the stochastic model is practically equivalent to sampling recipients from this group in the deterministic model, if a $q$ fraction of the group deterministically participate.

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
