[Supplementary Material]

# Appendix

## A  Notation Glossary

| | |
|---|---|
| **Sets of Agents** | |
| $N$ | Set of agents in the population |
| *Recipients* | Set of agents who receive invitation letters (random variable) |
| *Pool* | Set of agents in the pool (random variable) |
| *Panel* | Set of agents on the panel (random variable) |
| **Sortition Panel Parameters** | |
| $n$ | Size of the population |
| $r$ | Number of invitation letters sent out |
| $k$ | Size of the panel |
| $F$ | Set of all features |
| $V_f$ | Set of possible values for a specific feature $f \in F$ |
| $F(i)$ | Feature vector of agent $i$ |
| $n_{f,v}$ | Number of agents in the population with value $v$ of feature $f$ |
| $\ell_{f,v}, u_{f,v}$ | Lower and upper quotas for every feature-value pair |
| $q_i$ | Probability that agent $i \in N$ enters the pool, conditioned on being invited |
| $q^*$ | Minimum value of $q_i$ over all agents ($q^* := \min_{i \in N} q_i$) |
| $\alpha$ | Parameter defined as $\alpha := q^* r / k$ |

## B  Supplementary Material for Section 3

### B.1  Discussion of Theorem Preconditions

We show that pools are good with high probability under two preconditions: that each feature-value group constitutes at least $1/k$ fraction of the population (so $n_{f,v}/n \geq 1/k$ for all $f, v$), and that the number of recipients is sufficiently high relative to the participation probabilities and the panel size ($\alpha = q^* r / k \to \infty$).

The first condition is natural because if a group should proportionally receive less than one seat on the panel, any positive lower bound on selection probabilities for agents in groups would violate proportionality.

The second condition enforces that the number of agents invited $r$ is large enough relative to the minimum participation probability $q^*$ and the size of the panel. Without this condition, there can be a constant probability that the pool will feature zero agents with a certain feature-value: Suppose that $\alpha$ is an arbitrary positive constant, set all $q_i := \alpha k/r$, and consider a feature-value pair $f, v$ with $n_{f,v} = n/k$ agents. In expectation, there will be $(r/n)(n/k) = r/k$ agents with feature-value $f, v$ among the recipients. If $r \in \omega(k)$, there are at most $2 r/k$ such recipients with high probability. Then, the probability that the pool contains no agent with $f, v$ is at least

$$(1 - \alpha k/r)^{2 r/k} = (1 - q_i)^{2 \alpha/q_i} = \big( \underbrace{(1 - q_i)^{1/q_i}}_{\to\, 1/e \text{ as } q_i \to 0} \big)^{2\alpha} \to e^{-2\alpha} > 0.$$

### B.2  Discussion of Ties to Discrepancy Theory

In rounding agents' marginal selection probabilities to select a panel, we round fractional variables to 0 or 1 such that the sum of certain sets of variables changed only by a small amount. This problem is closely connected to *combinatorial discrepancy* [9, 23], which can be summarized in the same words, by additionally assuming that the initial fractional values are $1/2$. In fact, the original Beck-Fiala theorem arises in the context of discrepancy, showing that, if each variable appears in a bounded number $t$ of sets, discrepancy $\Theta(t)$ can be achieved (where in our setting, $t$ corresponds to $|F|$, the number of features). Beck and Fiala [3] conjectured that it is actually possible to achieve discrepancy in $\mathcal{O}(\sqrt{t})$. Should this conjecture be true, similar ideas might translate to our setting to guarantee the satisfaction of quotas closer to exact proportionality. To this day, however, the best known bound in $t$

is still in $\Theta(t)$ [7]. In accordance with this result, we guarantee a relaxation of $|F|$ from proportional representation of groups.

We note that there do exist other disrepancy results that give sub-linear dependencies on $|F|$, but at the cost of introducing dependencies on other parameters. One such result is Theorem 5.3 in [2], which guarantees discrepancy a square-root dependency on $|F|$. However, subject to our requirement that the per-person marginal probability must deviate from $k/n$ by only $\pm\delta k/n$ where $\delta \in o(1)$, Bansal's result guarantees a discrepancy bound of $O(\sqrt{|F|\log(kn/\delta)})$, which grows in $n$, making it unfavorable in our setting.

## B.3 Proof of Lemma 2

The results in this section allow $k \geq 1$ and $r \geq 1$ to vary arbitrarily in $n$; they just require that $\alpha := q^* r/k \to \infty$ as $n \to \infty$ (without requiring $\alpha$ to grow at a specific minimum rate relative to $n$). All convergences are relative to $n$ going to infinity.

**Lemma 2.** *Suppose that $\alpha \to \infty$ and $n_{f,v} \geq n/k$ for all $f, v$. Then, for all agents $i \in Population$, $\mathbb{P}[Pool \text{ is good} \mid i \in Pool] \to 1$.*

In the following proofs, it is convenient to refer to $1/q^*$, the largest possible value of $a_i$, as $a^*$. Note that $a^* = \frac{r}{\alpha k}$. We will refer to the random set of recipients with a certain feature-value pair $f, v$ as $Recipients_{f,v} := \{i \in Recipients \mid f(i) = v\}$.

We begin by showing in Lemmas 5 and 7 that, conditioned on $i$ being in the pool, the following three events occur with high probability:

**A.** $k \, a^* \leq \sum_{j \in Pool} a_j$

**B.** $\sum_{j \in Pool} a_j \in [(1 - \alpha^{-.492})\, r, (1 + \alpha^{-.492})\, r]$

**C.** $\sum_{j \in Pool: f(j)=v} a_j \in [(1 - \alpha^{-.492})\, \frac{n_{f,v}}{n}\, r, (1 + \alpha^{-.492})\, \frac{n_{f,v}}{n}\, r] \quad \forall f, v$

We then show in Lemma 8 that, when these events occur on some pool, the pool must be good, which concludes the proof of Lemma 2.

**Lemma 5.** *Under the assumptions of Lemma 2, $\mathbb{P}\left[\text{Event A} \wedge \text{Event B} \mid i \in Pool\right] \to 1$.*

*Proof.* Fix the set of recipients $R$ (including $i$). With respect to the randomness in the pool self-selection, the random variables $a_j \cdot \mathbb{1}\{j \in Pool\}$ across all $j \in R \setminus \{i\}$ are independent, bounded in $[0, a^*]$, and have expected value $a_j q_j = 1$. Thus, by a Chernoff bound, and using that $a^* = r/(\alpha k)$,

$$\mathbb{P}\left[\left|\sum_{j \in Pool \setminus \{i\}} a_j - (r-1)\right| \geq \alpha^{-.495}\, (r-1)\right] \leq 2\, e^{-\alpha^{-.99}\, \frac{r-1}{a^*}/3}$$

$$= 2\, e^{-\alpha^{-.99}\, \frac{r-1}{r}\, \alpha k/3}$$

$$\leq 2\, e^{-\Omega(\alpha^{.01})} \to 0,$$

where the last inequality uses the fact that $r \geq 2$ for large enough $n$[11] and that $k \geq 1$.

Conditioning on this high-probability event, it follows that, for large enough $n$,

$$\sum_{j \in Pool} a_j \geq 1 + \sum_{j \in Pool \setminus \{i\}} a_j \geq 1 + (1 - \alpha^{-.495})\, (r-1) \geq (1 - \alpha^{-.492})\, r,$$

which shows the lower bound in Event B. For the upper bound,

$$\sum_{j \in Pool} a_j \leq a^* + \sum_{j \in Pool \setminus \{i\}} a_j \leq a^* + (1 + \alpha^{-.495})\, (r-1) \leq r/(\alpha k) + (1 + \alpha^{-.495})\, r$$

$$\leq (1 + \alpha^{-.495} + 1/\alpha)\, r \leq (1 + \alpha^{-.492})\, r \leq 1/(1 - \alpha^{-.492})\, r.$$

This establishes Event B.

For large enough $n$, the lower bound on $\sum_{j \in Pool} a_j$ can be extended as

$$\sum_{j \in Pool} a_j \geq (1 - \alpha^{-.492})\, r \geq r/\alpha \geq k\, a^*,$$

which shows Event A. $\qquad\square$

For Event C, we need to show that $\sum_{j \in Pool: f(j)=v} a_j$ is concentrated for a feature-value pair $f, v$. As an intermediate step, we first show that the *number* of pool members ("$\sum_{j \in Pool: f(j)=v} 1$") with this feature-value pair is concentrated:

**Lemma 6.** *Under the assumptions of Lemma 2, for each $f, v$,*

$$\mathbb{P}\left[(1 - \alpha^{-.495})\,\frac{n_{f,v}}{n}\, r \leq \left|Recipients_{f,v}\right| \leq (1 + \alpha^{-.495})\,\frac{n_{f,v}}{n}\, r \,\middle|\, i \in Pool\right] \to 1.$$

*Proof.* Conditioned on $i \in Pool \subseteq Recipients$, $Recipients \setminus \{i\}$ is distributed as if $r - 1$ members of $Population \setminus \{i\}$ were drawn with equal probability and without replacement. Thus,

$$\mathbb{E}\left[\left|Recipients_{f,v}\right|\,\middle|\, i \in Pool\right] = \begin{cases} n_{f,v}\,\frac{r-1}{n-1} & \text{if } f(i) \neq v \\ 1 + (n_{f,v} - 1)\,\frac{r-1}{n-1} & \text{if } f(i) = v. \end{cases}$$

In both cases, we show that $\mathbb{E}\left[\left|Recipients_{f,v}\right|\,\middle|\, i \in Pool\right] \in \left[(1 - k/r)\, n_{f,v}\,\frac{r}{n},\, (1 + k/r)\, n_{f,v}\,\frac{r}{n}\right]$. Indeed, for the upper bound,

$$\mathbb{E}\left[\left|Recipients_{f,v}\right|\,\middle|\, i \in Pool\right] \leq 1 + (n_{f,v} - 1)\,\frac{r-1}{n-1} \leq 1 + n_{f,v}\,\frac{r}{n} = \left(1 + \frac{n}{n_{f,v}}/r\right) n_{f,v}\,\frac{r}{n}$$

$$\leq (1 + k/r)\, n_{f,v}\,\frac{r}{n} \leq (1 + 1/\alpha)\, n_{f,v}\,\frac{r}{n}.$$

For the lower bound,

$$\mathbb{E}\left[\left|Recipients_{f,v}\right|\,\middle|\, i \in Pool\right] \geq n_{f,v}\,\frac{r-1}{n-1} = \frac{r-1}{r}\, n_{f,v}\,\frac{r}{n} = (1 - 1/r)\, n_{f,v}\,\frac{r}{n}$$

$$\geq (1 - k/r)\, n_{f,v}\,\frac{r}{n} \geq (1 - 1/\alpha)\, n_{f,v}\,\frac{r}{n}.$$

As the (independent) union of the deterministic set $\{i\}$ and indicator variables for sampling without replacement, the variables $\mathbb{1}\{j \in Recipients\}$ satisfy negative association and therefore Chernoff inequalities [25]. Thus, for the upper tail bound,

$$\mathbb{P}\left[\left|Recipients_{f,v}\right| \geq (1 + \alpha^{-.497})\,(1 + 1/\alpha)\, n_{f,v}\,\frac{r}{n}\,\middle|\, i \in Pool\right] \leq e^{-\alpha^{-.994}\,(1+1/\alpha)\, n_{f,v}\,\frac{r}{n}/3}$$

$$\leq e^{-\alpha^{-.994}\, n_{f,v}\,\frac{r}{n}/3} \leq e^{-\alpha^{-.994}\,\frac{r}{k}/3} \leq e^{-\alpha^{-.994}\,\alpha/3} \leq e^{-\alpha^{.006}/3} \to 0.$$

Similarly, for the lower tail bound,

$$\mathbb{P}\left[\left|Recipients_{f,v}\right| \leq (1 - \alpha^{-.497})\,(1 - 1/\alpha)\, n_{f,v}\,\frac{r}{n}\,\middle|\, i \in Pool\right] \leq e^{-\alpha^{-.994}\,(1-1/\alpha)\, n_{f,v}\,\frac{r}{n}/2}$$

$$\overset{(\alpha \geq 3)}{\leq} e^{-\alpha^{-.994}\, n_{f,v}\,\frac{r}{n}/3} \leq e^{-\alpha^{.006}/3} \to 0.$$

The claim follows from observing that, for $r/k$ large enough,

$$(1 - \alpha^{-.497})\,(1 - 1/\alpha) \geq 1 - \alpha^{-.497} - \alpha^{-1} \geq 1 - \alpha^{-.495}$$

and

$$(1 + \alpha^{-.497})\,(1 + 1/\alpha) = 1 + \alpha^{-.497} + \alpha^{-1} + \alpha^{-1.497} \leq 1 - \alpha^{-.495}. \qquad\square$$

**Lemma 7.** *Under the assumptions of Lemma 2, $\mathbb{P}\left[\textbf{Event C}\,\middle|\, i \in Pool\right] \to 1$.*

*Proof.* Fix a single feature-value pair $f, v$. By Lemma 6, with high probability, the number of recipients $r_{f,v}$ with feature-value pair $f, v$ is in

$$\left[(1 - \alpha^{-.495})\, \frac{n_{f,v}}{n}\, r, (1 + \alpha^{-.495})\, \frac{n_{f,v}}{n}\, r\right].$$

Going forward, we will fix a set of recipients $R$, and we assume that $r_{f,v}$ indeed falls in this range. For large enough $n$, this implies that $r_{f,v}$ is positive. For ease of notation, we will implicitly condition on $i \in Pool$ and these high-probability events.

The self-selection process of agents with feature-value pair $f, v$ might look a bit different depending on whether $f(i) = v$. If $f(i) \neq v$, the self selection of agents with feature-value pair $f, v$ is independent from our knowledge about $i$ being in the pool. Thus, the random variable $\sum_{\substack{j \in Pool, \\ f(j)=v}} a_j$ is the sum of independent random variables $a_j \, \mathbb{1}\{j \in Pool\}$ for each $j \in R, f(j) = v$, where each variable is bounded in $[0, a^*]$ and has expectation 1. In particular, $\mathbb{E}\left[\sum_{\substack{j \in Pool, \\ f(j)=v}} a_j\right] = r_{f,v}$.

Else, if $f(i) \neq v$, $\sum_{\substack{j \in Pool, \\ f(j)=v}} a_j$ is still the sum of independent random variables $a_j \, \mathbb{1}\{j \in Pool\}$ and each variable is bounded in $[0, a^*]$. However, the specific variable $a_i \, \mathbb{1}\{i \in Pool\}$ is deterministically $a_i$ (all other variables still have expectation 1). Thus, $\mathbb{E}\left[\sum_{\substack{j \in Pool, \\ f(j)=v}} a_j\right] = r_{f,v} - 1 + a_i$.

$$r_{f,v} - 1 + a_i = \left(1 + \frac{a_i - 1}{r_{f,v}}\right) r_{f,v} \leq \left(1 + \frac{a^*}{r_{f,v}}\right) r_{f,v} \leq \left(1 + \frac{r/(\alpha\,k)}{(1 - \alpha^{-.495})\, r\, n_{f,v}/n}\right) r_{f,v}$$

$$\leq \left(1 + \frac{r/(\alpha\,k)}{(1 - \alpha^{-.495})\, r/k}\right) r_{f,v} = \left(1 + \frac{1}{(1 - \alpha^{-.495})\,\alpha}\right) r_{f,v}$$

$$\leq (1 + 2/\alpha)\, r_{f,v}. \qquad\qquad\qquad\qquad \text{(for } \alpha^{.495} \geq 2\text{)}$$

Thus, across both cases, the expectation $\mathbb{E}\left[\sum_{\substack{j \in Pool, \\ f(j)=v}} a_j\right]$ is at least $r_{f,v} \geq (1 - \alpha^{-.495})\, \frac{n_{f,v}}{n}\, r$ and at most $(1 + 2/\alpha)\, r_{f,v} \leq (1 + 2/\alpha)\, (1 + \alpha^{-.495})\, \frac{n_{f,v}}{n}\, r \leq (1 + \alpha^{-.493})\, \frac{n_{f,v}}{n}\, r$ for large $n$, and we can use Chernoff bounds.

For bounding the lower tail,

$$\mathbb{P}\left[\sum_{\substack{j \in Pool, \\ f(j)=v}} a_j \leq (1 - \alpha^{-.495})(1 - \alpha^{-.495})\, \frac{n_{f,v}}{n}\, r\right] \leq e^{-\alpha^{-.99}(1 - \alpha^{-.495})\, \frac{n_{f,v}}{n}\, r/(2\,a^*)}$$

$$\overset{(\alpha^{.495} \geq 3)}{\leq} e^{-\alpha^{-.99}\, \frac{n_{f,v}}{n}\, r/(3\,a^*)} = e^{-\alpha^{-.99}\, \frac{n_{f,v}}{n}\, r/(3\,r/(\alpha\,k))} \leq e^{-\alpha^{-.99}\, \frac{n_{f,v}}{n}\, \alpha\,k/3}$$

$$\leq e^{-\alpha^{-.99}\,\alpha/3}$$

$$\leq e^{-\alpha^{.01}/3} \to 0.$$

For bounding the upper tail,

$$\mathbb{P}\left[\sum_{\substack{j \in Pool, \\ f(j)=v}} a_j \geq (1 + \alpha^{-.495})(1 + \alpha^{-.493})\, \frac{n_{f,v}}{n}\, r\right] \leq e^{-\alpha^{-.99}(1 + \alpha^{-.493})\, \frac{n_{f,v}}{n}\, r/(3\,a^*)}$$

$$\leq e^{-\alpha^{-.99}\, \frac{n_{f,v}}{n}\, r/(3\,a^*)} = e^{-\alpha^{-.99}\, \frac{n_{f,v}}{n}\, \alpha\,k/3} \leq e^{-\alpha^{-.99}\,\alpha/3} \leq e^{-\alpha^{.01}/3} \to 0.$$

Note that, for large $n$, $(1 - \alpha^{-.495})(1 - \alpha^{-.495}) \geq 1 - 2\,\alpha^{-.495} \geq 1 - \alpha^{-.492}$. Similarly, $(1 + \alpha^{-.495})(1 + \alpha^{-.493}) \in 1 + \mathcal{O}(\alpha^{-.493})) \leq 1 + \alpha^{-.492}$.

This shows that, for each $f, v$, $(1 - \alpha^{-.492})\, \frac{n_{f,v}}{n}\, r \leq \sum_{\substack{j \in Pool, \\ f(j)=v}} a_j \leq (1 + \alpha^{-.492})\, \frac{n_{f,v}}{n}\, r$ with high probability. The claim follows by a union bound over all (finitely many) feature-value pairs. $\qquad \square$

**Lemma 8.** *For large enough $n$, if Events A, B, and C occur for a pool $P$, $P$ is good.*

*Proof.* Suppose that Events A, B, and C occur in a pool $P$.

**Condition (1):** $\forall j \in P.\ 0 \leq \pi_{j,P} \leq 1$. Clearly, $\pi_{j,P}$ is nonnegative, and Event A implies that $\pi_{j,P} = k\, a_j / \sum_{j' \in P} a_{j'} \leq k\, a^* / \sum_{j' \in P} a_{j'} \leq 1$.

**Condition (2):** $\forall f, v.\ (1 - \alpha^{-.49})\, k\, n_{f,v}/n \leq \sum_{j \in P: f(j)=v} \pi_{j,P} \leq (1 + \alpha^{-.49})\, k\, n_{f,v}/n$. Fix any feature-value pair $f, v$. Recall that, by Event B,

$$\sum_{j \in P} a_j \in [(1 - \alpha^{-.492})\, r, (1 + \alpha^{-.492})\, r],$$

and, by Event C,

$$\sum_{j \in P: f(j)=v} a_j \in [(1 - \alpha^{-.492})\, \frac{n_{f,v}}{n}\, r, (1 + \alpha^{-.492})\, \frac{n_{f,v}}{n}\, r].$$

Observe that, for any $x \in [0, 1/3]$,

$$\frac{1+x}{1-x} \leq \frac{1 + x + x\,(1 - 3\,x)}{1 - x} = \frac{1 + 2\,x - 3\,x^2}{1 - x} = 1 + 3\,x.$$

Then, if $n$ is large enough such that $\alpha^{-.492} \leq 1/3$, it follows that

$$\sum_{j \in P: f(j)=v} \pi_{j,P} = k\, \frac{\sum_{j \in P: f(j)=v} a_j}{\sum_{j \in P} a_j} \leq k\, \frac{(1 + \alpha^{-.492})\, \frac{n_{f,v}}{n}\, r}{(1 - \alpha^{-.492})\, r} \leq (1 + 3\,\alpha^{-.492})\, k\, \frac{n_{f,v}}{n}$$

$$\leq (1 + \alpha^{-.49})\, k\, \frac{n_{f,v}}{n}.$$

Next, observe that, for any $x$,

$$\frac{1-x}{1+x} \geq \frac{1 - x - 2\,x^2}{1+x} = 1 - 2\,x.$$

Thus,

$$\sum_{j \in P: f(j)=v} \pi_{j,P} = k\, \frac{\sum_{j \in P: f(j)=v} a_j}{\sum_{j \in P} a_j} \geq k\, \frac{(1 - \alpha^{-.492})\, \frac{n_{f,v}}{n}\, r}{(1 + \alpha^{-.492})\, r} \geq (1 - 2\,\alpha^{-.492})\, k\, \frac{n_{f,v}}{n}$$

$$\geq (1 - \alpha^{-.49})\, k\, \frac{n_{f,v}}{n}.$$

**Condition (3):** $\sum_{i \in P} a_i \leq r/(1 - \alpha^{-.49})$. This follows from Event B since $\sum_{j \in P} a_j \leq (1 + \alpha^{-.492})\, r \leq (1 + \alpha^{-.49})\, r = \frac{1 - \alpha^{-.98}}{1 - \alpha^{-.49}}\, r \leq r/(1 - \alpha^{-.49})$ for large enough $n$. $\qquad \square$

## B.4 Proof of Lemma 3

### B.4.1 Rounding the Linear Program Using Discrepancy Methods

In Part II of the algorithm, we need to implement the marginal probabilities $\pi_{i,P}$ from Part I by randomizing over panels of size $k$. Additionally, the panels produced by this procedure should guarantee that the number of panel members of a feature-value pair $(f, v)$ lies in a narrow interval around the proportional number of panel members $k\, n_{f,v}/n$. Technically, this corresponds to randomly rounding the fractional solution $x_i := \pi_{i,P}$ of an LP, such that afterwards all variables are 0 or 1, i.e., indicator variables for membership in a random panel.

Formally, we prove the following lemma:

**Lemma 3.** *There is a polynomial-time sampling algorithm that, given a good pool $P$, produces a random panel Panel such that (1) $\mathbb{P}[i \in Panel] = \pi_{i,P}$ for all $i \in P$, (2) $|Panel| = k$, and (3) $\sum_{i: f(i)=v} \pi_{i,P} - |F| \leq |\{i \in Panel \mid f(i) = v\}| \leq \sum_{i: f(i)=v} \pi_{i,P} + |F|$.*

To round the linear program, we use an iterative rounding procedure based on the famous Beck-Fiala theorem [3]. For ease of exposition, we first describe an algorithm for deterministic rounding and describe in the subsequent subsection how to turn it into a randomized rounding procedure. From here on, we drop the index "$P$" from the marginal probabilities $\pi_{i,P}$, both for ease of notation and to emphasize that the lemma applies to any set of marginal probabilities adding up to $k$ (such other marginals might arise, say, from clipping and rescaling the $\pi_{i,P}$ if some of them are greater than 1).

**Lemma 9.** *For a pool $P$, let $(\pi_i)_{i \in P}$ be any collection of variables in $[0,1]$ such that $\sum_{i \in P} \pi_i = k$. Then, we can efficiently compute a deterministic 0/1 rounding $(x_i)_{i \in P}$ such that $\sum_{i \in P} x_i = k$ and such that, for each feature-value pair $f, v$,*

$$\sum_{i \in P: f(i) = v} \pi_i - |F| \leq \sum_{i \in P: f(i) = v} x_i \leq \sum_{i \in P: f(i) = v} \pi_i + |F|.$$

*Proof.* We initialize $x_i \leftarrow \pi_{i,P}$, and the following inequalities are therefore satisfied:

$$\sum_{i \in P} x_i = k \tag{4}$$

$$\sum_{i \in P: f(i) = v} x_i = \sum_{i \in P: f(i) = v} \pi_{i,P} \qquad\qquad \forall f, v. \tag{5}$$

We then iteratively update the $x_i$ and maintain a set of equations that starts as the equations in Eqs. (4) and (5), but from which we will iteratively drop some equations of type (5). Throughout this process, we maintain that the $x_i$ satisfy all remaining (i.e., not dropped) equations and that $x_i \in [0,1]$ for all $i$. We call $x_i \in (0,1)$ *active*; once an $x_i$ stops being active, it stays at its value 0 or 1 to the end of the rounding. We continue our iterative process until no more active variables remain, at which point we return our 0/1 rounding.

Whenever the number of remaining equalities is lower than the number of active agents, the values $x_i$ for the active variables must be underdetermined by the equalities. More precisely, after considering all inactive $x_i$ as constants, the space of remaining $x_i$ that satisfies the remaining equalities forms an affine subspace of non-zero dimension. Since this subspace must intersect the boundary of the unit hypercube, there is a way of updating the $x_i$ such that all equalities are preserved, such that no inactive variable gets changed, and such that at least one additional variable becomes inactive (progress).[12]

Else, we know that the number of active agents $n'$ is at most the number of remaining equalities $m$. If $m = 1$, i.e., if Eq. (4) is the only remaining equation, there cannot be any active agents since Eq. (4) can only be satisfied if no $x_i$ or at least two $x_i$ are non-integer. Thus, in the following, $m \geq 2$. For any remaining equality of type (5) corresponding to some feature-value pair $f, v$, say that it *ranges over $t$ many active variables* if there are $t$ many active variables $x_i$ such that $f(i) = v$. Should any of the remaining constraints range over all $n'$ many active variables, then this constraint must be implied by constraint (4) and the values of the inactive variables. We can thus drop the redundant constraint without consequences (progress), and repeat the iterative process.

If none of these steps apply, we show that some constraint of type (5) ranges over at most $|F|$ active variables: Clearly, this is the case if $n' \leq |F|$, and furthermore if $n' = |F| + 1$ because we removed constraints of type (5) ranging over all active variables. If $n' > |F| + 1$, note that every active agent appears in at most $|F|$ many equations of type (5), at most one per feature. It follows that the total number of active agents summed up over all remaining equalities of this type is at most $n'|F| < n'|F| - (|F| + 1) + n' = (n' - 1)(|F| + 1) \leq (m - 1)(|F| + 1)$, which implies that one of the $m - 1$ equalities of type (5) ranges over less than $|F| + 1$ active variables. Drop all such equalities (progress) and repeat.

Since $n' + m$ decreases in every iteration, this algorithm will produce a deterministic panel in polynomial time. Since constraint (4) is never dropped, the panel size must be exactly $k$. By how much might the equations of type (5) for a feature-value pair $f, v$ be violated in the result? Clearly, they are maintained exactly up to the point where they are dropped.[13] From this point on, however,

only $|F|$ many active variables could still change the value of $\sum_{i \in P : f(i) = v} x_i$. Since each of these variables remains in its range $[0, 1]$ throughout the rounding process, the final $x_i$ must satisfy

$$\sum_{i \in P : f(i) = v} \pi_i - |F| \leq \sum_{i \in P : f(i) = v} x_i \leq \sum_{i \in P : f(i) = v} \pi_i + |F|. \qquad \square$$

### B.4.2 Randomizing the Beck-Fiala rounding

We give two methods of transforming the previous deterministic rounding algorithm into a randomized rounding algorithm. To prove Lemma 3, we can directly apply a result by Bansal [2] to our deterministic rounding procedure:

**Lemma 3.** *There is a polynomial-time sampling algorithm that, given a good pool $P$, produces a random panel $Panel$ such that (1) $\mathbb{P}[i \in Panel] = \pi_{i,P}$ for all $i \in P$, (2) $|Panel| = k$, and (3) $\sum_{i : f(i) = v} \pi_{i,P} - |F| \leq |\{i \in Panel \mid f(i) = v\}| \leq \sum_{i : f(i) = v} \pi_{i,P} + |F|$.*

*Proof.* We apply Theorem 1.2 by Bansal [2] to the deterministic rounding procedure of Lemma 9. To apply the theorem, we need to give a $\delta > 0$ such that, when there are $n'$ many active variables left, the number of remaining equalities in the next iteration is at most $(1 - \delta) n'$ constraints. In Lemma 9, we showed that $m$ is always set to a value of at most $n' - 1$. Thus, for $\delta := 1/n$, we get that $m \leq n' - 1 = (1 - 1/n') n' \leq (1 - 1/n) n'$ and can apply the theorem. $\qquad \square$

While the previous algorithm runs in polynomial time, we found an alternative way of randomizing the rounding to be more efficient in practice. This technique is based on naïve column generation, which is not guaranteed to run in polynomial time, but has the following advantages:

- it uses linear programs rather than semi-definite programs,
- instead of a single random panel, the column generation (deterministically) generates a *distribution* over panels, which allows us to analyze the distribution after a single run, and
- there is a continuous progress measure that allows us to stop the optimization process once we implement the $\pi_i$ with sufficient accuracy.

We describe this algorithm in the proof of the following version of Lemma 3, which does not require polynomially-bounded runtime:

**Lemma 10.** *There is a sampling algorithm that, given a good pool $P$, produces a random panel $Panel$ such that (1) $\mathbb{P}[i \in Panel] = \pi_{i,P}$ for all $i \in P$, (2) $|Panel| = k$, and (3) $\sum_{i : f(i) = v} \pi_{i,P} - |F| \leq |\{i \in Panel \mid f(i) = v\}| \leq \sum_{i : f(i) = v} \pi_{i,P} + |F|$.*

*Proof.* First, note that we can strengthen Lemma 9 slightly by giving it an arbitrary vector $\vec{c} \in \mathbb{R}^{|P|}$ as part of its input and additionally requiring that $\langle \vec{c}, \vec{x} \rangle \geq \langle \vec{c}, \vec{\pi} \rangle$, where $\vec{x}$ is the vector of $x_i$ and $\vec{\pi}$ the vector of $\pi_i$. This stronger statement follows from the same proof if we require every update of the $x_i$ to additionally maintain that $\langle \vec{c}, \vec{x} \rangle \geq \langle \vec{c}, \vec{\pi} \rangle$. Since this intersects the non-zero dimensional affine subspace formed by the constraints with a half space that contains at least the current point $\vec{x}$, the resulting intersection is still unbounded, which means that we can find an intersection with the boundary of the hypercube. We refer to this procedure as the "modified Lemma 9."

Now, let $\mathfrak{B} \neq \emptyset$ be any set of panels satisfying the constraints of the lemma, possibly exponentially many. Consider the following linear program and its (simplified) dual:

PRIMAL($\mathfrak{B}$):

$$\textit{minimize } \delta$$

$$\textit{s.t. } \left| \pi_i - \sum_{B \in \mathfrak{B} : i \in B} \lambda_B \right| \leq \delta \quad \forall i \in P$$

$$\sum_{B \in \mathfrak{B}} \lambda_B = 1$$

$$\delta \geq 0, \lambda_B \geq 0 \quad \forall B \in \mathfrak{B}$$

DUAL($\mathfrak{B}$):

$$\textit{maximize } \left( \sum_{i \in P} \pi_i z_i \right) - \hat{z}$$

$$\textit{s.t. } \sum_{i \in B} z_i \leq \hat{z} \quad \forall B \in \mathfrak{B}$$

$$|z_i| \leq 1 \quad \forall i \in P$$

The primal LP searches for a distribution over the panels $\mathfrak{B}$ such that the largest absolute deviation between the marginal $\sum_{B\in\mathfrak{B}:i\in B} \lambda_B$ and the target value $\pi_i$ of any $i \in P$ is as small as possible. Let $\overline{\mathfrak{B}}$ denote the set of panels that can be returned by the modified Lemma 9, for any vector $\vec{c}$ in its input.

**Observation 1: For any $\mathfrak{B} \neq \emptyset$, the LP has an objective value $obj(\mathfrak{B}) \geq 0$.** Indeed, in the primal, the objective value is clearly bounded below by 0, and the LP is feasible for any distribution over $\mathfrak{B}$ and large enough $\delta$. By strong duality, the dual LP must have the same objective value.

**Observation 2: $obj(\overline{\mathfrak{B}}) = 0$.** For the sake of contradiction, suppose that the objective value was strictly positive, i.e., that $\vec{\pi}$ does not lie in the convex hull of $\overline{\mathfrak{B}}$. Then, there must be a plane separating $\vec{\pi}$ from this convex hull, and an orthogonal vector $\vec{c}$ such that $\langle \vec{c}, \vec{\pi} \rangle > \langle \vec{c}, \vec{x} \rangle$ for any $\vec{x}$ corresponding to a panel in $\overline{\mathfrak{B}}$. Applying the modified Lemma 9 with this vector $\vec{c}$ would lead to a contradiction.

Consider Algorithm 1, which iteratively generates a subset $\mathfrak{B} \subseteq \overline{B}$ by column generation.

---

**Algorithm 1:** Column generation

1   $\mathfrak{B} \leftarrow \{\text{result of running modified Lemma 9 with arbitrary } \vec{c}\}$
2   **while** $obj(\mathfrak{B}) > 0$ **do**
3      fix optimal values $z_i, \hat{z}$ for DUAL($\mathfrak{B}$)
4      $B \leftarrow$ result of running modified Lemma 9 with $\vec{c}$ as the vector of $z_i$
5      $\mathfrak{B} \leftarrow \mathfrak{B} \cup \{B\}$
6   **return** $\mathfrak{B}$

---

**Observation 3: Algorithm 1 terminates.** It suffices to show that, in Line 4, the generated panel $B$ is not yet contained in $\mathfrak{B}$ since, then, the size of $\mathfrak{B}$ grows in every iteration and is always upper-bounded by the finite cardinality of $\overline{\mathfrak{B}}$. By the definition of the modified Lemma 9, $B$ always satisfies $\sum_{i\in B} z_i \geq \sum_{i\in P} \pi_i z_i$. However, since the objective value is positive, any $B' \in \mathfrak{B}$ satisfies $\sum_{i\in P} \pi_i z_i > \hat{z} \geq \sum_{i\in B'} z_i$, which shows that $B \notin \mathfrak{B}$.

Once Algorithm 1 terminates with a set $\mathfrak{B}$, we know that $obj(\mathfrak{B}) = 0$, which means that, by solving PRIMAL($\mathfrak{B}$), we obtain a distribution over valid panels that implements the marginals $\pi_i$, which concludes the proof. $\qquad\square$

In practice, it makes sense to exit the while loop in Line 2 already when $obj(\mathfrak{B})$ is smaller than some small positive constant, which guarantees a close approximation to the marginal probabilities while reducing running time and preventing issues due to rounding errors.

## B.5   Proof of Theorem 1

**Theorem 1.** *Suppose that $\alpha \to \infty$ and $n_{f,v} \geq n/k$ for all feature-value pairs $f, v$. Consider a sampling algorithm that, on a good pool, selects a random panel, $Panel$, via the randomized version of Lemma 3, and else does not return a panel. This process satisfies, for all $i$ in the population, that*

$$\mathbb{P}[i \in Panel] \geq (1 - o(1)) \, k/n.$$

*All panels produced by this process satisfy the quotas $\ell_{f,v} := (1 - \alpha^{-.49}) \, k \, n_{f,v}/n - |F|$ and $u_{f,v} := (1 + \alpha^{-.49}) \, k \, n_{f,v}/n + |F|$ for all feature-value pairs $f, v$.*

*Proof.* The claim about the quotas immediately follows from Lemma 3 and the definition of a good pool. Concerning the selection probabilities,

$$\mathbb{P}[i \in Panel] = \sum_{\substack{\text{good pools } P \\ i \in P}} \mathbb{P}[i \in Panel \mid Pool = P] \, \mathbb{P}[Pool = P] = \sum_{\substack{\text{good pools } P \\ i \in P}} \frac{k \, a_i}{\sum_{j\in P} a_j} \, \mathbb{P}[Pool = P].$$

Since $\sum_{j \in P} a_j \leq r/(1 - \alpha^{-.49})$ for good pools, we continue

$$\geq (1 - \alpha^{-.49}) \, k/(r \, q_i) \underbrace{\sum_{\substack{\text{good pools } P \\ i \in P}} \mathbb{P}[Pool = P]}_{} = (1 - \alpha^{-.49}) \frac{k}{r \, q_i} \, \mathbb{P}[i \in Pool \wedge Pool \text{ is good}]$$

$$= (1 - \alpha^{-.49}) \frac{k}{r \, q_i} \underbrace{\mathbb{P}[Pool \text{ is good} \mid i \in Pool]}_{\in \, 1 \, - \, o(1) \text{ by Lemma } 2} \underbrace{\mathbb{P}[i \in Pool]}_{= q_i \, r/n} \in (1 - o(1)) \frac{k}{n}. \qquad \square$$

## C  Supplementary Material for Section 4

**Participation Model**   Let $y_i = 1$ for agents who would join the pool if invited, and $y_i = 0$ for agents who would not. We want to predict $q_i = \mathbb{P}[y_i = 1]$ for all agents in the pool. To do so, we learn the following parametric model, which describes the relationship between an agent's feature vector $F(i)$ and value of $q_i$.

$$q_i = \beta_0 \prod_{f \in F} \beta_{f, f(i)}$$

This type of generative model describes a decision process known as *simple independent action* [11, as cited in [28]]. To express this model in a more standard form, let $x_i$ be a vector describing agent $i$'s values for all features in $F$, where each index $j$ of $x_i$ corresponds to a feature-value $f, v$ and contains a binary indicator of whether agent $i$ has value $v$ for feature $f$. Let $M$ be the length of $x_i$, where $M = 1 + \#feature\text{-}values$. We then reshape parameters $\beta_0$, $\beta_{f,v}$ for all $f, v$ into a parameter vector $\boldsymbol{\beta}$ of length $M$, and correspondingly, $x_i$ must have value 1 at its first index for all agents $i$, corresponding to the parameter $\beta_0$. We can then write an equivalent version of our model in more standard form. Note that $q_i$ is technically a function of $x_i, \boldsymbol{\beta}$, but we omit this notation for simplicity.

$$q_i = \prod_{j \in [M]} \boldsymbol{\beta}_j^{x_{i,j}}$$

**Maximum Likelihood Estimation with Contaminated Controls**   To estimate the parameters $\boldsymbol{\beta}$ of this model on fixed pool $P$ and fixed background sample $B$, we apply the estimation methods in Section 3 of Lancaster and Imbens [16]. We use the objective function in Equation 3.3, which is designed to perform maximum-likelihood estimation (MLE) in the setting of contaminated controls. Let $z_i$ be an indicator such that $z_i = 1$ for $i \in P$ and $z_i = 0$ for $i \in B$. Let $w_i$ be the weight of agent $i \in B$ (for details on these weights, see Appendix D). Recall that $\overline{q}$ is the average participation probability in the underlying population. Then, the likelihood function $L(\boldsymbol{\beta})$ that we would maximize to directly learn our model is

$$L(\boldsymbol{\beta}) = \sum_{i \in B \cup P} \left( z_i \sum_{j \in [M]} \left( x_{i,j} \log \boldsymbol{\beta}_j \right) - w_i \log \left( \overline{q} \, |B|/|P| + \prod_{j \in [M]} \boldsymbol{\beta}_j^{x_{i,j}} \right) \right)$$

Unfortunately, $L(\boldsymbol{\beta})$ is not obviously concave in $\boldsymbol{\beta}$. To get around this, we re-parameterize our model such that we can instead learn the *logarithms* of our parameters. Defining a new parameter vector $\theta$ such that $\theta_j = \log(\boldsymbol{\beta}_j)$ for all $j \in [M]$, we can rewrite our model equivalently as the exponential model.

$$q_i = \prod_{j \in [M]} \boldsymbol{\beta}_j^{x_{i,j}} = \exp\left( \log \left( \prod_{j \in [M]} \boldsymbol{\beta}_j^{x_{i,j}} \right) \right) = \exp\left( \sum_{j \in [M]} x_{i,j} \log(\boldsymbol{\beta_j}) \right) = e^{\theta x_i}$$

By Equation 3.3 in Lancaster and Imbens [16], the likelihood function $L'(\theta)$ we maximize is now the following. By Theorem 11, this objective function is concave, so it can therefore be maximized efficiently (under the constraint that $\theta \leq 0$).

$$L'(\theta) = \sum_i \left( z_i \theta x_i - w_i \log \left( \overline{q} \, |B|/|P| + e^{\theta x_i} \right) \right) \qquad (6)$$

**Theorem 11.** *The log-likelihood function for the simple independent action model under contaminated controls is concave in the model parameters.*

*Proof.* The first term of the sum is linear, so both concave and convex. The second term is concave by Lemma 12. □

**Lemma 12.** *Let function $f(\theta) = -\log(c + e^{\theta X})$, where $c > 0$ is a constant. $f$ is concave.*

*Proof.* The $i,j$th term of the Hessian matrix $H$ of $f$ can be written as

$$H_{i,j} = -X_i X_j \frac{ce^{\theta X}}{(c + e^{\theta X})^2}$$

Now, let $\psi = \frac{\sqrt{ce^{\theta X}}}{c + e^{\theta X}}$. Noting that $X$ is considered a column vector, we can then rewrite the Hessian in terms of $\psi$ as $H = -(\psi X)(\psi X)^T$. In words, the negative Hessian can be written as the outer product of the vector $\psi X$ with itself. Therefore, the negative Hessian is positive semi-definite, and the Hessian is negative semi-definite, implying that $f$ is concave. □

**Discussion of Methods**   The reader may note that we treat $\bar{q}$ as a known constant in our estimation, but the objective function we use from Lancaster and Imbens is designed for the setting in which $\bar{q}$ is a variable. There is precedent in the literature for doing so [27]. As Lancaster and Imbens discuss, using $\bar{q}$ as a constant rather than a variable when maximizing Equation 3.3 introduces issues of over-parameterization, because it is not enforced that the average $q_i$ over the population be $\bar{q}$. While we cannot estimate $q_i$ values for the entire population for lack of data, it would be a worrying sign if the average $q_i$ over the *background sample*, a uniform sample from the population, was far from our assumed $\bar{q}$. However, we find that the average of our estimated $q_i$ values over the background sample is 2.9%, which matches $\bar{q} = 2.9\%$.

# D   Supplementary Material for Section 5

For estimation, we use two datasets. For our positively-labeled data, we use the set of pool members from the UK Climate Assembly (for details, see Appendix D.1). For our background sample, we use the European Social Survey (ESS), which serves as an unlabeled uniform sample of the population.

## D.1   Climate Assembly UK Details & Pool Dataset

Our pool dataset contains the agents from the pool of the *Climate Assembly UK*, a national-level sortition panel on climate change held in the UK in 2020. We use "panel" to refer to the group of people who deliberate, and "assembly" to refer to the actual deliberation step. The panel for this assembly as selected by the *Sortition Foundation*, a UK-based nonprofit that selects sortition panels. A document by the Sortition Foundation gives the following description of this assembly:[14]

> *This Citizens' Assembly will meet across four weekends in early 2020 to consider how the UK can meet the Government's legally binding target to reduce greenhouse gas emissions to net zero by 2050. The outcomes will be presented to six select committees of the UK parliament, who will form detailed plans on how to implement the assembly's recommendations. These plans will be debated in the House of Commons.*

In the formation of the panel for this assembly, 30 000 letters were sent out inviting people to participate. Of these letter recipients, 1 727 people entered the pool, and 110 people were selected for the panel. The features and corresponding sets of values used for this panel are described in Table 1.

Those with value *Other* for gender were dropped from the pool data because an equivalent value could not be constructed in the ESS data. This resulted in us dropping 12 people out of the original 1727, for a pool dataset of final size 1715. Note that dropping these people did not affect our estimate of $\bar{q}$ — before and after dropping these agents, it was 2.9%. The Climate Concern Level feature was dropped altogether from the set of features used for analysis because there were too few people in the pool with value *Not at all concerned* to give these agents proportional representation on the panel.

| Feature ($f \in F$) | Values ($V_f$) |
|---|---|
| Gender | Male, Female, ~~Other~~ |
| Age | 16-29, 30-44, 45-59, 60+ |
| Region | North East, North West, Yorkshire and the Humber, East Midlands, West Midlands, East of England, London, South East, South West, Wales, Scotland, Northern Ireland |
| Education Level | No Qualifications/Level 1, Level 2/Level 3/Apprenticeship/Other, Level 4 and above |
| ~~Climate Concern Level~~ | ~~Very concerned, Fairly concerned, Not very concerned, Not at all concerned, Other~~ |
| Ethnicity | White, Black or ethnic minority (BAME) |
| Urban / Rural | Urban, Rural |

Table 1: Climate Assembly UK features and values.

Due to privacy agreements between the Sortition Foundation and the pool members, we are unable to share this dataset.

## D.2 Background Data

We define the size of the ESS dataset to be the sum of the weights of the agents within it.[15] For details on weights, see the *Re-weighting* paragraph of this section. In order to use this data as our background sample, we construct feature vectors for each person in the ESS data that correspond to those used in Climate Assembly UK, as defined in Table 1.

In this section, we describe how we constructed the variables corresponding to the features and their values as specified by the Sortition Foundation. We dropped 44 people out of the original 1959 people in the ESS dataset, and we briefly discuss this decision and its implications. Finally, we describe how we re-weighted the ESS data to correct for sampling and non-response bias to approximate the scenario in which the surveyed individuals were uniformly sampled from the population. This step is important because, in our $q_i$ estimation procedure, we assume that our background sample is uniformly sampled.

**Variable construction**  Fortunately, the ESS data contained variables and categories that either exactly or very closely corresponded to the features and values specified by the Sortition Foundation. Essentially the only modification to the ESS data we made to construct valid feature vectors was the aggregation over categories in the *Education Level* and *Urban/Rural* ESS variables, which were broken down into more fine-grained categories than those specified in Table 1. In general, for features with values containing the value "other", missing data was assigned the value "other". Below is a table showing which variables and values from the ESS data were used to construct each feature from the Climate Assembly UK. Exact details on how these variables were used is documented in the code (see Appendix D.3 for reference to readme).

| Feature (Climate Assembly UK) | Variable (ESS raw data) |
|---|---|
| Gender | gndr |
| Age | agea |
| Region | region |
| Education Level | edulvlb |
| Climate Concern Level | wrclmch |
| Ethnicity | blgetmg |
| Urban/Rural | domicil |

**Dropping people**  As described in Table 1, the Climate Assembly's youngest valid age category was 16-29. We therefore dropped all four people in the ESS data who were under 16 years old.

Dropping people who fall outside our demographic ranges of interest is not a problem for weights, because the weights of all people of interest (who we want to be fair to) will remain the same relative to each other, and we care only about the composition of this relevant population.

There were an additional 40 people who may have been within our demographic range of interest, but who were missing age, race, or urban/rural data. Among these 40 people, 33, 6, and 4 people did not have data for variables corresponding to the features *age*, *ethnicity*, and *urban / rural*, respectively. While dropping these people could affect the weighting scheme, the distribution of weights of those dropped is strongly right-skewed, meaning that those who we dropped belong to groups that tended to be oversampled in the ESS data. These people are therefore likely more numerous in the ESS data overall, and dropping some of them will have a smaller proportional effect.

Finally, the ESS did not permit people to answer "other" for gender, a category permitted on the Sortition Panel. Without any way to construct the *gender = other* feature-value in the ESS data, we dropped the members of the Climate Assembly pool with this feature-value.

**Re-weighting**    The ESS recommends re-weighting their data to correct for bias, and they provide multiple sets of possible weighting schemes for doing so[16]. Of the provided options, we elected to apply the Post-Stratification Weights, because these weights account for not only sampling bias, but also non-response bias, by incorporating auxiliary information from other demographic surveys. By this weighting scheme, each person in the ESS data is given a weight $w_i$, representing how much that person should count in the analysis of the ESS data, where the weights are normalized to 1. This weight is encoded in the ESS data as 'pspwght'.

**Estimation of** $\overline{q}$    We bolster the identification of our model with an estimate of $\overline{q}$, the rate of true positives in the population. In our setting, this is the number of people who would ultimately enter the pool if invited. We estimate $\overline{q}$ in Climate Assembly UK data roughly as the fraction of people who joined the pool (1 715) out of those who were invited (30 000). These numbers seem to imply that the $\overline{q} \approx 1\,715/30\,000 = 5.7\%$. However, there is a complication: each letter is sent to a *household*, rather than an individual, and any eligible member of an invited household may join the pool. Using the ESS data, we compute (see below) the average number of eligible panel participants per household to be 2.00, implying that in reality, 60 000 eligible people were invited to participate in the pool. As a result, we estimate $\overline{q}$ to be $\overline{q} = 1\,715/60\,000 \approx 2.9\%$.

Let $ESS$ be the set of agents in the cleaned ESS data. Computing the average number of eligible panel participants per household from the ESS data is not entirely trivial, because sampling *people* uniformly (or in the case of the ESS, approximating uniform sampling by re-weighting) is biased toward larger households. To account for this, for each person $i \in ESS$, we scale their weight $w_i$ by the inverse of the number of eligible people in their household, $householdsize_i$. Then,

$$\text{average number of eligible people per household} = \frac{\sum_{i \in ESS} \left( \frac{w_i}{householdsize_i} \right) \cdot householdsize_i}{\sum_{i \in ESS} \left( \frac{w_i}{householdsize_i} \right)}$$

We compute $householdsize_i$ for each person $i \in ESS$ using the weighted ESS data. Age is the only feature from the UK Climate Assembly for which the ESS data may contain values rendering a person ineligible (specifically, the ESS data surveys people down to age 15, while the climate assembly accepted only those over 16). To count the number of people in each household who are eligible, we use variables 'agea', 'pspwght', and 'yrbrn2-12', which describe the ages of person $i$'s household members (up to 12 household members).

### D.3    Implementation Details

Our experiments were implemented in Python, using PyTorch for the MLE estimation and Gurobi for solving the linear programs in the column generation. Our code is contained in the supplementary material and will be made available as open source when published. The file "README.md" in the code gives detailed instructions for reproducibility.

We found the log-likelihood presented in Eq. (6) to be easy to maximize. For accuracy, we chose a small step size of $10^{-5}$ and a large number $10^5$ of optimization steps. The final objective was 4157.32345, and objective changes between iterations 20 000 and 100 000 were less than $3 \times 10^{-6}$.

Our experiments were run on a 13-inch MacBook Pro (2017) with a 3.1 GHz Dual-Core i5 processor. Optimizing the log-likelihood took 46 seconds. Running the column generation took 38 minutes to reach the desired accuracy of $10^{-6}$, which is much smaller than the smallest $\pi_{i,P}$ at around $2\%$. For the version including climate concern, MLE estimation took 37 seconds reaching a log-likelihood of 4601.01427, and column generation took 26 minutes.

Sampling 100 000 pools each and simulating our algorithm for the end-to-end experiments took 30 minutes for $r = 10\,000$, 55 minutes for $r = 11\,000$, 61 minutes for $r = 12\,000$, 76 minutes for $r = 15\,000$, and 95 minutes for $r = 60\,000$. All running times should be seen as upper bounds since other processes were running simultaneously. Sampling the same number of pools for the case including the climate concern feature took around 410 minutes for $r = 600\,000$. The equivalent experiments with the greedy algorithm took around 19 hours (floor and ceiling quotas) and around 12 hours (no quotas).

### D.4 Results and Validation of $\beta, q_i$ Estimation

**Pool and Background Data Composition**  First, we examine the frequency at which each feature-value occurs in the pool and the background data. As shown in the figure below, those with the most education are highly over-represented in the Climate Assembly UK pool compared to the background sample, and people with low education are under-represented. Similarly, we see men are slightly over-represented in the pool, and increasing age also seems to increase likelihood of entering the pool.

**Estimates of $\beta$**  We find that $\beta_0 = 8.8\%$, meaning that all agents participate with a baseline probability of $8.8\%$. In the figure below are estimates of $\beta_{f,v}$ for all feature-values $f, v$. Recall that $1 - \beta_{f,v}$ can be interpreted as the probability of not participating due to having value $v$ for feature $f$; in other words if $\beta_{f,v}$ is 1, then feature-value $f, v$ has no adverse effect on whether a person participates.

Notably, these $\beta$ estimates are consistent with the composition of the pool compared to the background data. For example, people of increasing age were increasingly over-represented in the pool compared to the background data, and we see here that $\beta$ associated with age increase with increasing age. Similarly, we see that having low education greatly diminishes a person's likelihood of participation, corresponding to the observation that the pool contained a disproportionately low number of people with the two lower levels of education. In fact, one can confirm that across all feature-values, $\beta$ values correspond with the composition of the pool data compared to the background data, indicating that the $\beta$ values learned with our model are a good fit to the data used to learn them.

**Gender**
Male — 100
Female — 83

**Age**
16-29 — 68
30-44 — 74
45-59 — 88
60+ — 100

**Education**
Level 1 and below — 38
Level 2-3 or other — 50
Level 4 and above — 100

**Urban/Rural**
Urban — 100
Rural — 76

**Region**
North East — 67
North West — 64
Yorkshire/Humber — 68
East Midlands — 78
West Midlands — 83
East of England — 86
London — 69
South East — 79
South West — 100
Wales — 81
Scotland — 65
Northern Ireland — 79

**Ethnicity**
White — 84
BAME — 100

**Estimates of $q_i$** We compute our $q_i$ estimates based on $\beta$ estimates according to the model in Appendix C. We get the following distributions of $q_i$ values in the pool and background datasets.

The data shown in this plot is limited to density of $q_i$ values between 1% and 8%, because bins outside this range contain fewer than 7 people, and are withheld to avoid potential privacy issues. Less than 0.3% of agents in either dataset are excluded for this reason.

Not very surprisingly, we find that the pool overrepresents agents with higher participation probability with respect to their share in the background sample.

**Test for Calibration of $q_i$ Estimates** To validate whether our model fits the data well, we form a *hypothetical pool* by imagining that the weighted background sample was selected as the set of recipients and that the members of this set participate with our estimated probability $q_i$. For some attributes that agents might have or not have, the expected number of agents in the hypothetical pool with this attribute is

$$\sum_{i \in B : i \text{ has attribute}} q_i.^{17}$$

Since the set of invitation recipients to the Climate Assembly and the background sample are both assumed to be representative samples of the population, we would expect the above sum to be (close to) proportional to the fraction of pool members with this attribute — at least if the model fits the data well.

For instance, this idea allows us to re-examine the previous plot of $q_i$ values by letting the orange bars not denote the (scaled) *number* of members in the background sample with $q_i$ in the right range, but instead the (scaled) *sum of $q_i$ values* of members in the background sample with $q_i$ in this range.

The fact that these distributions align fairly well can be seen as our $q_i$ passing a sort of calibration test — of those agents with a certain $q_i$ value, roughly a $q_i$ proportion would participate when invited. Relative to our background sample, the Climate Assembly pool does not seem to untypically skew towards agents with low or high values of $q_i$.

Once again, for privacy reasons we display frequencies of $q_i$ values only between 1% and 8%. Once again, less than $0.3\%$ of agents in either dataset are excluded for this reason.

**Comparison of Realized Pool Composition and Hypothetical Pool Composition**    We now plot the same comparison between the Climate Assembly pool and the hypothetical pool but for the prevalence of each feature-value pair.

The figure below shows that if our $\beta$ estimates and the $q_i$ estimates they yield are true for members of the population, then if we sampled the underlying population as was done to form the Climate Assembly UK pool, we would get in expectation a pool that looks almost identical to realized pool. This illustrates in another way that our $\beta$ estimates are a good fit to the data we provided.

**Testing Model Capture of 2-Correlations**    Our model assumes that each feature-value affects people's probability of participating independently of all other feature-values. This analysis tests whether this causes our model to severely misjudge the participation probability for some group defined by the intersection of *two* feature-value pairs, again comparing the prevalence of these groups in the Climate Assembly UK pool vs. the hypothetical pool that would be drawn from a population

with the same composition as the background sample. On the plot below, each point represents an intersection of two feature-values. Each point's $x$ and $y$ coordinates are the fraction of people with that intersection in the Climate Assembly UK pool and the fraction of the hypothetical pool, respectively. We would hope for this relationship to be exactly linear, illustrating that each pair of feature-values occurs at the same rate in the real vs. hypothetical pool.

### D.5 Details on End-To-End Experiment

As described in the body of the paper, we generate a synthetic population by scaling up the ESS participants to a population of 60 million individuals. The number of copies of a participant is proportional to their weight in the ESS, and is rounded to an integer using the Hamilton apportionment method. 100 000 times per experiment, we select a set of letter recipients of size $r$ uniformly from the population, and flip a biased coin with probability $q_i$ for each letter recipient to determine whether she joins the pool. For each pool, we then obtain the selection probabilities of the pool members conditioned on this being the pool (or an unbiased estimate of these probabilities):

- For our algorithm, we check whether the pool $P$ is good. If the pool is not good, we (conservatively) assume that no panel is returned, and that pool members have zero probability of being selected. Else, we return the selection probabilities $\pi_{i,P}$.
- We use the implementation of the greedy algorithm developed by the Sortition Foundation and available at https://github.com/sortitionfoundation/stratification-app/tree/4a957359b708a327aad0103ab2a59d061aeaeeb4.
  Since we do not have a closed form for individual selection probabilities, we run the greedy algorithm 10 times and report the average time that each pool member was selected. While these estimates of selection probabilities are noisy, they are unbiased estimates of the end-to-end probability and independent between pools. Thus, the noise largely averages out over the 100 000 random pools. In no case did the greedy algorithm fail to satisfy the quotas.

Each point in the diagrams corresponds to one agent in the ESS sample and indicates this agents' $q_i$ as well as the average selection probability of its copies, averaged over the different pools and the different copies. Since both our algorithm and the greedy algorithm treat agents with equal feature vector symmetrically, averaging over the copies of an ESS participant is a valid way to estimate the end-to-end probability of any single copy, which greatly reduces sample variance.

In the body of the paper, we mention the behavior of the greedy algorithm without any quotas. In this case, the panel members seem to be sampled with near-equal probability from the pool, which leads to end-to-end probabilities that are roughly proportional to $q_i$:

## D.6 Additional Results for Section 5

**End-to-End Fairness Results for Varied $r$ Values**  This plot shows the end-to-end probabilities for all agents in the synthetically-generated population over varied values of $r$. To recall, we copied the agents in the background sample (in proportion to their weight) to obtain a synthetic population of size 60 million (the order of magnitude of eligible participants for the Climate Assembly).

We display these end-to-end probabilities for $r$ values 11 000, 12 000, 13 000, and 60 000, where 60 000 is the $r$ value used to form the real-life Climate Assembly UK pool. Every point in the scatter plot corresponds to an original member of the background sample, and the point's y-value is the mean selection probabilities averaged over 100 000 sampled pools and over all copies of this background agent.[18]

An important question is what we do when a bad pool occurs. In the corresponding figure in the body of the text (examining only $r = 60\,000$), we did not credit any selection probability to any agent when bad pools occurred. When we take this approach for multiple $r$ values, the result shows a sharp discontinuity between $r = 11\,000$ (when everyone's end-to-end probability is essentially zero) and $r = 12\,000$ (when it is around 95%). As it turns out, the property that makes nearly all pools bad when $r = 11\,000$ is Eq. (3). Note that this property is the least consequential of the three defining properties of a good pool: if we proceed with Part II of the algorithm on a pool that satisfies only Eqs. (1) and (2), we still satisfy the quotas but just can't bound the end-to-end probabilities. Since the end-to-end probabilities are what we are measuring here anyway, we will in the following graph count bad pools as good pools if they only violate Eq. (3).

As shown in the figure below, we see a smooth transition towards the end-to-end guarantee, where higher values of $r$ give better guarantees. The agents with the lowest selection probabilities are suffering most from low values of $r$, with their end-to-end probability trailing that of the majority of other agents. From $r = 15\,000$ upwards, however, all agents in the population receive an end-to-end probability that is very close to $k/n$. This threshold roughly coincides with the point at which $\alpha$ becomes larger than one.

## D.7 Validation and Results including Climate Concern Feature

This section includes all the analysis in this paper and appendices, re-done with the climate concern level feature included. Figures in this section are provided in the same order as they were presented in the body of the paper, Appendix D.4, and Appendix D.6.

We omit the figure showing end-to-end probabilities at $r = 60,000$, because when the *Climate Concern Level* feature is included, good pools are so rare at this value of $r$ that all end-to-end probabilities are 0. Similarly, for the greedy algorithm, the floor and ceiling quotas are often not satisfiable. In 754 out of 1 000 random pools, this is because fewer pool members are "not at all concerned" about climate change than the lower quota for this feature, which is 6. In 86 out of the remaining pools, the greedy algorithm fails to identify a valid panel within the first 100 restarts. Only in the remaining 160 pools did the greedy algorithm find a valid panel in fewer than 100 iterations.

**Pool and Background Data Composition**

**Estimates of $\beta$**

**Estimates of $q_i$** $\quad \beta_0 = 24.3\%$. Frequencies of $q_i$ values above 15% are not shown due to privacy concerns. 6.8%, 1% of agents in pool, background datasets respectively are not presented for this reason.

**Test for calibration of $q_i$ estimates**    Frequencies of $q_i$ values above 20% are not shown due to privacy concerns. Less than 0.4% of agents in either dataset are not presented for this reason.

**Comparison of Realized Pool Composition and Hypothetical Pool Composition**

**Testing model capture of 2-correlations**

*(Figures from Appendix D.6)*

**End-to-End Fairness Results for Varied $r$ Values**    This figure demonstrates that, for large enough $r$, we can get $k/n$ end-to-end probability for all agents in the synthetic population when we include the Climate Concern Level feature. We only include analysis for only one $r$ value because the $r$ values must be extremely large to give any end-to-end guarantees when the Climate Concern Feature is included, and running the analysis with such large $r$ costs substantial computational time.

## Footnotes

[11]Since $r = \alpha k/q^* \geq \alpha/q^* \geq \alpha \to \infty$.

[12]This step can be implemented in polynomial time by solving systems of linear equations.

[13]We do not count if the equality was dropped because it was implied by constraint (4), in which case it is preserved exactly throughout the rounding.

[14]https://docs.google.com/spreadsheets/d/1kwgOpxMX4pwR3Myu4pXku4gjcnOS53bPOKwOGjZNxyI/edit#gid=0

[15] This sum should ideally be equal to the number of people in the ESS data, but because we drop a few people, the sum of weights no longer exactly equals the number of people.

[16]https://www.europeansocialsurvey.org/docs/methodology/ESS_weighting_data_1.pdf

[17]Of course, all operations on the background sample respect the weights, which we ignore here for the sake of clarity.

[18]Averaging over the copies of an agent makes use of the fact that the selection process treats copies of the same agent symmetrically, which makes the empirical means converge faster.