[Reviews · NeurIPS 2020]

Review 1

Summary and Contributions: The paper studies the problem of selecting a sortation panel of size k among n agents, in settings in which different agents may have different willingness to participate. This can make uniform sampling biased and unfair, when invited agents in different populations agree to participate in the panel at different rates. The main contributions of the paper are as follows: 1) Assuming knowledge of the participation probability for each agent, the authors provide a sampling algorithm with the following fairness guarantees: i) every agent has a probability close to k/n to be selected, and ii) the distribution of feature-value pairs in the panel accurately reproduces that of the original population. I.e., everyone has a fair shot at being part of the label, and feature-value pairs are proportionally represented. The guarantees hold provided that the pool of invited agent that replied affirmatively is good. 2) The authors show that as the parameter \alpha (and in turn, the number of invited agents r) grows, the probability of pool being good goes to 1, and the guarantees of point 1 apply. 3) The authors show how to learn the probabilities of participation when they are not known, under an assumption on the function form of said probabilities. 4) Finally, the authors provide experimental result to illustrate and complement the theory.

Strengths: One of the main strength of the paper is that it seems to give strong fairness guarantees. Theorem 1 shows that the main algorithm guarantees that each agent has a probability close to k/n to be part of the panel, and that the panel satisfies fairly strict quotas on the number of certain feature-value pairs present in the panel. The theorem holds under some conditions, but these conditions are reasonable. On the one hand, one needs $n_{f,v} \geq n/k$ (or to ignore the feature-value pairs that do not satisfy this condition) for the problem to make sense in the first place. Otherwise, it may be the case that (f,v) appears so infrequently in the population that assigning even a single slot on the panel of (f,v) would break proportionality. On the other hand, \alpha needs to be large; otherwise, q^* may be too small to guarantee a panel of size k, or r might be too small to get a sufficiently good pool to sample from. Further, the experiments nicely complement these results. They show that the lower and upper quotas obtained by their algorithm in practice tend to be very accurate, and significantly better than those predicted in theory. In turn, their algorithm is practical for ranges of \alpha that are larger than those predicted by theory, which is a valuable lesson for a decision-maker who would want to use the techniques developed in this paper in real life. Finally, the paper gives a complete picture of the problem at hand, by also providing a way to estimate the participation probabilities. There, the authors go beyond the naïve setting in which one has access to a representative sample of the feature-value pairs in the population, and instead consider a more realistic limited feedback setting in which one can only observe the features of agents who decide to participate.

Weaknesses: One weakness of the paper is that the convergence results are not very fine-grained, in the sense that part of those results are stated for \alpha going to infinity, but not for finite \alpha as is the case in practice. As such, it is hard to understand from the main body of the paper how changing the value of \alpha affects some of the guarantees of the algorithm, and how big \alpha needs to be in these theoretical guarantees. This seems to be a problem in the exposition itself rather than in the techniques, as the appendix proof seem to in fact explicitly compute dependencies on \alpha. I also think that the learning part of the paper is significantly weaker than the algorithmic and experimental ones. First, the results there seem to rely on a modified version of maximum likelihood estimation that comes from previous work; the authors’ contribution seems to be limited to using a fairly straightforward re-parametrization to obtain a concave likelihood function, making it efficiently optimizable. Second, the authors provide no theoretical guarantees on how well they recover the participation probabilities. Can any such guarantees be proved?

Correctness: The idea behind the proofs of the paper and its claims seem reasonable, however I have not checked for correctness of the calculations.

Clarity: The paper is well written and easy to follow. The model and panel selection process are clearly laid out, and the story of the paper and how the different parts fit together is clear. I also like that the authors provide a discussion of the conditions they need for their theorem statement to hold, though I think the paper would have gain from having those in the main body rather than the appendix.

Relation to Prior Work: To the best of my knowledge, the related work does an appropriate job of discussing the related work on each of the main topics it covers, i.e. individual and group-level fairness, self-selection bias, and sortition. They also clearly make the point that the novelty of their paper in the context of sortation comes from the additional of self-selection bias considerations that were not present in previous work that assumes perfect participation.

Reproducibility: Yes

Additional Feedback: Post author feedback comments: thanks to the authors for their response! The authors' comments about non-asymptotic bounds make sense, and reinforce my opinion that this paper should be accepted.


Review 2

Summary and Contributions: The paper deals with sortition—the selection of political officials as a random sample from a larger pool of candidates. The authors claim that uniform selection from the population is infeasible, as in practice only a fraction of the agents are willing to participate. The latter becomes a severe obstacle since the willingness to participate often correlates with individual attributes, such as gender, education, or race, resulting in a self-selection bias. Consequently, current sortition practices fail to represent individual segments accurately. The authors build on the existing sortition practice (reaching to recipients to create a pool, and then sampling a panel) and propose a selection algorithm that de-biases the self-selection effect. Their algorithm relies on de-biasing each agent’s marginal probability to join the panel and then employing an existing rounding technique to make sure several fairness properties are satisfied. The authors show that as long as some quantity (alpha, which depends on the willingness to attend, the recipient pool, and the panel sizes) goes to infinity, their algorithm satisfies desirable properties. More practically, they relax the full-information assumption on the willingness to join the panel and show these values can be estimated from data using a naïve model and evaluate their methods on synthetic data extrapolated from a real-world dataset.

Strengths: * The paper addresses a real-world problem that could benefit from a more systematic (or even axiomatic) approach. * The paper is exceptionally well-written. The authors propose a comprehensive background discussion and explain their methods in a crystal-clear manner. *The proposed solution is neat, easy to implement, and relatively efficient. The authors conduct experiments that complement their theoretical analysis and evaluate their algorithm for non-asymptotic input, showing satisfying results.

Weaknesses: * Some of the modeling assumptions lack proper justification (see questions below). * The infinite-time analysis (solely) of the algorithm weakness the technical contribution, which is claimed to solve a real-world solution.

Correctness: To the best my knowledge.

Clarity: Exceptionally well-written.

Relation to Prior Work: Yes.

Reproducibility: Yes

Additional Feedback: Overall, I enjoyed reading this paper. However, I have several concerns about the model etc. that I hope the authors could address in their rebuttal (especially 1-4). 1. The “end-to-end fairness” desideratum is a bit hard to take. The q quantity is used in this paper to neutralize bias, but it can also be viewed as agents valuation for a seat at the panel. Recall that every agent i with q_i>0 is selected to the panel with probability k/n, regardless of q_i. If we think of q_i as the extent to which agent i values a seat at the panel (a scarce resource in the fair division terminology), the authors require that agents get items regardless of their valuation. How is the random allocation fair to individuals? If anything, I would imagine requiring a “fair” rule to be weakly increasing in q_i and not inversely (while not damaging the representation of sub-populations). 2. The algorithm’s guarantees are valid when alpha goes to infinity, but in realistic scenarios, alpha will be much smaller (in the experiments section, alpha = 4.25). What is the barrier to providing finite-sample guarantees? If I understand correctly, the infinity assumption is needed for witnessing every feature-value (f-v) pair in a pool with high probability. However, I believe that a pair f-v abstaining from the pool is well-justified in case its portion is extremely low (the authors filtered out a pair of f-v in their experiment because it was not represented in the background data). If we are willing to accept an additive quota (allowing zero representatives of f-v if it is rare) instead of a multiplicative one, could the authors craft the finite-sample guarantees of their algorithm? If not, what is the main obstacle? 3. The authors do not present the problem formally, only the current solution as the baseline. In particular, the authors state steps 1,2, and 3 as part of the problem, while this structure might be limiting in the first place. For instance, we might benefit from a more focused targeting (sending letters to recipients in a non-uniform manner). Can we also modify step 1 and prevent littering people with 30,000 letters? (as the authors report in Line 283). If the authors had modeled the panel selection problem without forcing the existing structure (of steps 1,2, and 3), they might have found better methods. 4. Relating to the previous point, the results of the experiments are hard to interpret due to the lack of baselines. The authors could use the existing approach to contrast the results of their algorithm and show its superiority. In the current write-up, I cannot tell whether the methods are even required since the representation (or lack thereof) of uniform sampling is not examined. 5. There are several difficulties with the authors’ definition of fair representation. First, the authors propose quotas for f-v pairs, but gerrymandering issues can arise. For example, say we have two features with two values each (e.g., man and female, white and black). A panel of 50% white males and 50% black females is highly unrepresentative but satisfies the authors’ requirements. I admit that this issue is not unique to this paper, but typically in the fairness literature, one is after 2-3 minority groups and not numerous f-v pairs; hence, this problem is exacerbated in the paper. Second, as the authors say in Line 367, “the fairness of the process hinges on the entire machine-learning pipeline.” The generative model for estimating q_i ignores the knowledge collected in the “fairness in classification” literature. The joining probability of an individual depends on her gender, color, age, etc. To me, this sounds like discrimination. I’m not claiming that I have a better way of estimating q_i, but that actively using these estimates might be perceived problematic. Could the authors offer a remedy? Anyway, the authors should better address this point in their broader impact section. 6. Regarding the Beck-Fiala theorem: the authors might already be aware of this point, but to be on the safe side. Banaszczyk [1] proposes a version of the theorem that might be better for the problem at hand (see theorem 3 in [2] for a crystal-clear statement of the result). In short, this theorem offers a discrepancy of \sqrt{F log (FV)}, where F is the number of features, and V is the maximal number of values per feature (max_{f\in F} |V_f| in the terminology of the paper). Perhaps the authors could use this theorem to improve or augment their results? Additional comment: * The data for the experiments is private and will not be published with the paper. This might be disappointing to some of the readers. The authors could have stated that explicitly in their paper, instead of commenting on that in their README file. * Line 808 – “the the” [1] Wojciech Banaszczyk. Balancing vectors and Gaussian measures of n-dimensional convex bodies. Random Structures Algorithms, 12(4):351–360, 1998. [2] http://www.cs.toronto.edu/~anikolov/Prague18/Lectures/lect3.pdf ___ Post rebuttal notes: I thank the authors for their elaborated answers to my questions, some of which are indeed illuminating. I modified my overall score to 6 accordingly.


Review 3

Summary and Contributions: The paper introduces an algorithm for assembling a sortition panel that (with high probability) satisfies: End-to-End Fairness: The probabilities of each individual in the population to participate in the panel are asymptotically identical. Deterministic Quota Satisfaction: The selected panel satisfies certain upper and lower quotas enforcing approximate representation for a set of specified features. Computational Efficiency: Works in polynomial time. The claimed contribution here, comparing to other practices typically in use, is the fairness property and the computational efficiency (current methods only apply heuristics). The sortition process follows through the steps below: - r invitation letters are sent to a random subset of the population. - The ones that respond affirmatively form a 'pool' of agents. - A non-uniform sampling algorithm chooses the k participants of the panel from the pool. The paper focuses on constructing the algorithm in step 3 such that all properties mentioned above are satisfied. The end-to-end fairness is achieved in the following manner. For each sub-group of the population, the participation rate (probability that an agent responds affirmatively to the invitation) q_i is computed by comparing the prevalence of that group in the pool with general prevalence in the population (which is assumed to be known from a background sample. It also implicitly assumes that the $r$ randomly selected letter recipients reflect the population well in terms of the representation of each group, but that is not in the focus of the paper). Then, a distribution over all possible panels of size k is constructed, such that the marginal of each agent $\pi_i$ is proportional to $1/q_i$. The algorithm also guarantees the 'Deterministic Quota Satisfaction'. The algorithm only works for what the authors refer to as 'good pools', i.e. pools that satisfy some conditions that in general assure that the $\pi_i$ assignment make sense. They show that for r goes to infinity the probability of a good pool goes to 1. ************* I thank the authors for their answers to my questions

Strengths: from the 7 papers in my review pool, this is the most likely to be useful in practice. results are convincing, and may offer some useful tools and concepts to sortition designers.

Weaknesses: claimed contribution relies quite essentially on its advantages over greedy algorithms, in terms of end-to-end fairness and efficiency. While the efficiency advantage is clear, I think the authors should elaborate more on the first aspect. To me, it is not at all clear why sampling randomly from each subgroup in the pool separately until quotas are fulfilled, for example, yields less fairness. It is suggested that the introduced algorithm "guarantees to individuals safeguard against systematic under-representation of demographic groups unprotected by quotas" (line 360). it is important to note that the groups have to identified and their size in the population has to be estimated in order for this to happen. The discussion seems to be making an obvious point (that an agent is always more likely to get selected if she participates deterministically) but perhaps I'm missing something.

Correctness: The simulations are convincing although it would have been nice to have more than one dataset.

Clarity: The paper is very well written

Relation to Prior Work: It is not clear however where exactly is the technical contribution of the paper. Results are relaying on two powerful results from the literature: one for sampling with guaranteed marginals [3], and one from contaminated control [16]. This is great but still unclear if using these results is straightforward, or do they require all kinds of nontrivial conditions that have to be proved. From the appendix, the use of [3] seems straighforward although the authors expand on how to achieve a randomized rounding. The use of [16] seems to require some more effort. Another approach to fix uneven sampling is by using delegation, see e.g. Cohensius, Gal, et al. "Proxy Voting for Better Outcomes." AAMAS'2017 although here the bias is only due to a too-small sample (in the limit sortition is fine).

Reproducibility: Yes

Additional Feedback: - I think the statement of Thm. 1 (190) need not be this early in the paper - 'it' line 32 - 'k' line 83 - 'inequalities' line 560, appendix


Review 4

Summary and Contributions: The paper shows how to remove self-selection bias in a natural setting of biased two-stage sampling motivated by selecting sortition panels (but I guess that there must be other natural applications as well). Given a population of size n, we want to select a panel of size k so that each group (or "feature") of the population is almost equally represented in the panel. That would be easily solved by random sampling, but each person i may deny to join the panel with probability 1-q_i (mostly determined by the person's "features", but impossible to estimate for the entire population). So, in practice, we select a large biased pool of size r (by extending invitations to an appropriately large fraction of the population). The problem considered in the paper is how to select an unbiased panel of size k from such a pool of size r. The problem obviously calls for methods and techniques from Discrepancy Theory (see [9]) and this is what the paper actually does. The paper shows that under some reasonable assumptions on the composition of the pool and under the rather strong and unrealistic assumption that the biases q_i are known for the members of the pool, the straightforward approach of selecting each member i of the pool to the panel with probability proportional to k / q_i guarantees almost equal representation of each "feature" and that each member of the population could end up in the panel with probability approximately k/n (fairness). The proof of this claim is simple and follows from a series of Chernoff bound applications. To actually select the panel, the paper applies the Beck-Fiala iterative rounding approach to the probabilities above combined with a recent strong result of Bansal. This is the most technically interesting part of the paper. Again, there is not much novelty in this result. But it requires a knowledgeable application of deep and beautiful techniques. I found the assumption that the biases q_i of the pool members are known to the algorithm borderline unrealistic. The paper presents a rather simplistic model according to which these probabilities are generated. Learning these probabilities from this model can be performed by a straightforward application of MLE. On the positive side, I found the experimental evaluation interesting and convincing.

Strengths: -- The problem of removing self-selection bias when selecting sortition panels is natural, interesting, very well motivated and timely. If accepted, the paper would most likely attract more work on the topic. -- The rounding part of the algorithm is interesting, in the sense that deriving it requires a knowledgeable application of deep and beautiful techniques. -- The paper (main part) is extremely well written. It is a very enjoyable reading. -- The experimental evaluation is interesting and convincing.

Weaknesses: -- Lack of technical novelty and technical depth (compared against the average NeurIPS paper, with the exception of the rounding part of the algorithm). The assignment of marginals is literally the first thing that one would think about. The approach to learning the biases q_i is elementary and the generative model is rather simplistic. -- The problem is very similar to the most standard problems considered in the field of Discrepancy Theory. The approach in sec. 3.2 is hardly surprising. -- I found the assumption that the biases q_i of the pool members are known to the algorithm borderline unrealistic. This is a crucial assumption, the entire paper is based on it. I was not sold on the claim that "we can estimate these q_i from data available to practitioners".

Correctness: Yes, to the best of my technical ability and understanding.

Clarity: The paper (main part) is extremely well written. It is a very enjoyable reading. The problem of selecting a sortition panel in a fair and representative way is very well introduced and motivated. The contributions, the approach, the main results and the experimental evaluation are clearly described.

Relation to Prior Work: Rather yes. I would expect a more detailed overview of the problems, the techniques and some basic related results from the field of Discrepancy Theory.

Reproducibility: Yes

Additional Feedback: -- In Theorem 1, I suggest that you should provide some intuition (in the main part of the paper) about the reason that |F| is part of the quota bound. May be the same for \alpha^{0.49}, but this one is easy to guess. Post author feedback comments: the response did not change my opinion about the paper. I am still in favor of acceptance.

[Author Response · NeurIPS 2020]

We thank the reviewers for their insightful comments. In addition to the changes described below, we will make clearer
in the paper our contributions, algorithmic guarantees, and the conditions required for applying specific results.

**Reviewer 1 (R1)**
*1. Non-asymptotic guarantees*. R1 is correct that our proofs immediately translate to non-asymptotic bounds: In
Appendix B, we bound all error probabilities by simple terms that depend only on $\alpha$; simply adding up these terms (for
their union bound) produces results for finite (but large enough for some inequalities to hold) $\alpha$. For practical instances
with low $\alpha$, however, bounding dependencies in $n, r, k, q*$ and the $n_{f,v}$ only in terms of $\alpha$ can be lossy, and (as R2
rightfully points out) one would want to loosen quotas on some groups in exchange for lower failure probability. The
only obstacle in getting bounds for general $n, r, k, q^*, n_{f,v}$ are the messy algebraic dependencies that don't make for a
nice general formula. If the values of these parameters are specific numbers, it is easy to extract much sharper bounds
from our proofs. ($*$) We will discuss these points in the paper.

**Reviewer 2 (R2)** (for discussion of point #2, see response to R1 #1)
*1. End-to-end guarantee is unfair when $q_i$ is a valuation*. Foremost, we do not consider $q_i$ as purely an agent's valuation
for a panel seat, but rather as also capturing their *ability* to join the panel. Constraints on participation ability are
documented in a survey by Jacquet[1], and include scheduling conflicts, social anxiety, and family/work. Secondly, the
rule R2 proposes (one weakly increasing in $q_i$) is fundamentally incompatible with creating proportional panels. To see
why, suppose men and women are split 50/50 in the population, but women have low $q_i = 1/4$, and men have high
$q_i = 3/4$. Then, the pool is likely split about 25/75. If our algorithm sampled women with lower probability than men,
women would comprise less than $1/4$ of the panel in expectation, while their proportional share is $1/2$.

*3. Explored solutions limited to existing selection procedure*. Our modeling choices are dictated by how panel selection
is (and in many ways, must be) done in practice. In particular, forming a pool (step 2) is required due to limited
participation, and the process of sampling letter recipients (step 1) is constrained by the fact that practitioners usually
do not have detailed individual-level demographic population data, so systematically oversampling subpopulations that
participate at lower rates is not practicable. This said, more complicated sampling methods that *do not* require such data
(e.g., sending multiple rounds of letters) could worth exploring as an improvement on the status quo.

*4. No baseline comparison in experiments*. There are two reasonable baselines: uniform sampling, and the currently-
used greedy algorithm. Uniformly sampling the pool will simply give each agent $i$ an end-to-end probability proportional
to $q_i$ (up to negligible differences), since that the probability of them entering the pool is $r/n \, q_i$, so this baseline doesn't
warrant experiments. We have done experiments showing that the Sortition Foundation's greedy algorithm gives
substantially worse individual fairness guarantees than our algorithm, and ($*$) we will add these results to the paper.

*5. Definition of fair representation (gerrymandering, prediction on sensitive covariates)*. It is true that group fairness
guarantees (quotas) *alone* do not address gerrymandering concerns; this is a main reason that existing algorithms,
which only guarantee quotas, can be very unfair. At least in terms of expected representation, our algorithm cannot
have gerrymandering issues: since all agents' selection probabilities are $\approx k/n$, *any* subset of the population will be
represented near-proportionally by linearity of expectation. To prevent gerrymandering in ex-post representation, we
have seen practitioners use the cross product of features (in R2's example, gender and race) as a single feature. To R2's
point about $q_i$ prediction methods being perceived as discrimination: in the fairness in classification literature, using
protected attributes to counteract inequality is an accepted practice. That said, the concern about errors in the $q_i$s being
seen as discrimination is relevant; ($*$) we will discuss this and all points made here in the paper/broader impact section.

*6. Additional Beck-Fiala Results*. Thanks for suggesting Banaszczyk's result: an algorithmic version is in [2,Thm 5.3].
However, we want the per-person marginal probability to deviate by only $\pm \delta k/n$, and this would incur a discrepancy of
$O(\sqrt{|F| \log(n/\delta k)})$, which now depends on $n$. (We confirmed this with the author of [2].) ($*$) That said, we are happy
to mention this related result.

**Reviewer 3 (R3)** (for discussion of point #2 (clarification on individual guarantees) see response to R2 #5)
*1. Unclear why greedy is less fair*. At a high level, the greedy algorithm is unfair because it permits gerrymandering
(see R2 #5 and our response). In more detail, sampling each subgroup in greedy fashion (e.g. according to which quotas
are furthest from filled) can result in people of certain groups having near-zero probability of being on the panel. ($*$) We
can provide a simple example in the manuscript illustrating this problem, as well as experimental results (see R2 # 4).

**Reviewer 5 (R5)**
*1. Not sold on the claim that $q_i$ values are known or can be estimated*. The reviewer is correct not to take on faith that
the $q_i$s are known by the algorithm; this is why, in our experiments, we show they can be estimated. We explicitly
address the estimation of $q_i$ values in lines 296–304. Additionally, in Appendix D.4, we provide several pieces of
experimental evidence showing that, using data available to practitioners, we can estimate $q_i$ values that fit the data well.

## Footnotes

[1]Vincent Jacquet. Explaining Non-Participation in Deliberative Mini-Publics (2017). Eur J Polit Res 56.3 640–659.


[Meta-Review · NeurIPS 2020]

The reviewers had several concerns, though they felt that merits of the paper outweigh drawbacks overall, making this a borderline paper.